# Downscaling satellite-derived soil moisture in the Three North region using ensemble machine learning and multiple-source knowledge integration

Ka Liu[1,2], Hongyan Zhang[1,3], Yong Bo[1,3], Dehui Li[1,3], Long Li[1,3], Hang Li[1], Shudong Wang[1,2], Xueke Li[4]

[1]State Key Laboratory of Remote Sensing Science, Aerospace Information Research Institute, Chinese Academy of Sciences, Beijing, China
[2]Collaborative Innovation Center on Forecast and Evaluation of Meteorological Disasters (CICFEMD), Nanjing University o Information Science & Technology, Nanjing, China
[3]University of Chinese Academy Science, Beijing, China
[4]Department of Earth and Environmental Science, University of Pennsylvania, Philadelphia, Pennsylvania, 19104, USA

*Correspondence to*: Shudong Wang (wangsd@aricas.ac.cn) and Xueke Li (xuekeli@sas.upenn.edu)

**Abstract.** Soil moisture plays a crucial role in hydrological and ecological systems. While remote sensing has advanced large-scale soil moisture monitoring, current satellite products often face spatial resolution limitations. This study presents a reliable framework for downscaling satellite-derived soil moisture, leveraging ensemble machine learning and multiple knowledge sources. Our approach efficiently converges outputs from diverse machine learning algorithms through Bayesian model, harnessing spatiotemporal domains and point-wise data. Covering approximately five million square kilometres in the Three Northern region of China, our model generates 1-km daily soil moisture maps, accurately reflecting soil water content patterns and showing spatial consistency with outputs from two credible numerical models. Validation against in situ measurements from three ground networks confirms the accuracy of the downscaled dataset. Comparative analysis demonstrates the superiority of the Bayesian-based method over four individual machine learning methods. The high-resolution dataset produced proves effective in capturing drought dynamics, particularly extreme drought patterns. The robustness of our framework is further affirmed through uncertainty analysis, employing leave-one-out and progressive sample reduction approaches. In summary, our ensemble machine learning-based framework offers an efficient solution for acquiring accurate and high-resolution soil moisture data across large regions, with implications for water resource management and drought monitoring.

## 1 Introduction

Soil moisture serves as a nexus between surface water and groundwater, playing a foundational role in land surface ecosystem and hydrological cycle. It influences essential hydrological processes such as water infiltration, surface runoff, soil evaporation, and vegetation transpiration. Soil moisture monitoring has become pervasive across various domains, including applications in drought surveillance (Zhang et al., 2016), crop yield estimation (Chen et al., 2011), weather

forecasting (Drusch et al., 2009), etc. Considering its importance, the Global Climate Observing System (GCOS) and the United Nations Framework Convention on Climate Change (UNFCCC) have recognized soil moisture as a fundamental variable within terrestrial realms. Reliable soil moisture datasets are crucial but urgently demanded for research and applications in hydrology (Robinson et al., 2008; Western et al., 2004), agriculture, climatology (Anderson et al., 2007) and water resources management (Bastiaanssen et al., 2000; Dobriyal et al., 2012).

Various manners and methodologies have been designed for obtaining soil moisture. In-situ measurements offer temporal dynamics at the station level through extensive regional, national, and global monitoring networks (Dorigo et al., 2011; Ochsner et al., 2013). However, achieving spatial continuity across expansive regions remains challenging due to the limited ground stations. Advances in remote sensing technology, such as microwave sensors, have enabled the collection of soil moisture data on a larger scale (Petropoulos et al., 2015). Given their susceptibility to soil moisture variations and reduced susceptibility to diverse weather conditions, microwave remote sensors exhibit the capability to accomplish all-weather earth observation. A diverse of satellite-based soil moisture products is readily accessible, including Advanced Microwave Scanning Radiometer-Earth Observation System (AMSR-E) (Njoku et al., 2003), Soil Moisture Active Passive (SMAP) (Entekhabi et al., 2010), Soil Moisture and Ocean Salinity (SMOS) (Berger et al., 2002), and the Fengyun (FY) series of satellites developed independently by the National Satellite Meteorological Center of China Meteorological Administration (Parinussa et al., 2014). To obtain optimal spatial and temporal coverage and generate long duration sequence of soil moisture data, as part of the Climate Change Initiative (CCI), the European Space Agency (ESA) produces the combined microwave soil moisture product, ESA CCI SM (Dorigo et al., 2017). While these products are valuable for certain applications (Molero et al., 2016), the spatial resolution of these products—largely tens of kilometers—limits the ability to capture he spatial heterogeneity of soil moisture (Njoku and Entekhabi, 1996; Schmugge, 1998).

Soil moisture downscaling, an effective technique for improving spatial resolution, has received substantial attention (Zhang et al., 2022). Statistical approaches and land surface models (Famiglietti et al., 2008; Grayson and Western, 1998) have been widely used, but these methods typically require large amounts of parametric data with ground data. Various fusion methods integrating multi-source satellite remote sensing data have been developed, falling into categorized like active-passive microwave and optical-microwave data integration. While active-passive microwave fusion offers high accuracy, its cost and data processing complexity are notable. All these mentioned models encounter challenges related to model structure constraints, data quality, scale disparities, and geographic limitations (Peng et al., 2017; Werbylo and Niemann, 2014).

With rapid advancements in computer performance and artificial intelligence, the integration of machine learning techniques into soil moisture downscaling has marked a stride forward (Ali et al., 2015). Techniques like random forest (RF) and support vector regression (SVR) (Abbaszadeh et al., 2019; Srivastava et al., 2013) have showed the potential to achieve high accuracy and resolution in soil moisture data. Nonetheless, current studies often rely on a single machine learning method or the comparison of disparate single methods, inadvertently exposing themselves to limitations related to the heterogeneous nature of environmental processes (Jordan and Mitchell, 2015). Deep learning, a new generation technology in machine learning, shows promise across various soil moisture studies (Fang et al., 2019; Liu et al., 2022a; Ma et al., 2024). However,

this complex architectural model requires substantial training data to capture complicated patterns compared to traditional machine learning, posing challenges when applied to satellite-derived soil moisture data with sample limitations. Meanwhile, deep learning models may face difficulties in generalizing effectively across diverse environmental conditions, potentially degrading performance in regions not adequately represented in the training data (Ma et al., 2021b).

The utilization of ensemble machine learning emerges as an imperative alternative, combing the strengths of various learners to enhance robustness and uncertainty assessment. While a few studies, such as Shangguan et al. (2023) and Abbaszadeh et al. (2019), have combined downscaling results from multiple machine learning techniques to improve soil moisture downscaling accuracy, ensemble machine learning in soil moisture downscaling encounters challenges related to uncertainties. These challenges include selecting the optimal blend of models, harmonizing outputs from models trained on distinct datasets, and assigning weights to optimize performance (Ma et al., 2021a; Sagi and Rokach, 2018; Zounemat-Kermani et al., 2021). Existing ensemble machine learning often overlooks the incorporation of prior knowledge, a crucial regularization mechanism that prevents overfitting and enhances model generalization. This may magnify systematic errors and biases among individual models instead of correcting them. Collectively, the extent to which ensemble machine learning contributes to improving downscaled soil moisture quality remains unclear.

Given the limitations of current soil moisture downscaling methods, we propose a robust ensemble machine learning framework that considers both spatial and temporal domains, with a focus on prior knowledge from in-situ ground measurements. Covering the extensive Three Northern region in China (over five million square kilometres), our study addresses the challenges posed by diverse landscapes and climatic variations. Our work assesses the model's robustness and effectiveness through: 1) evaluating downscaled results using diverse in-situ measurements; 2) comparing with alternative machine learning methods and credible land surface model outputs; 3) demonstrating the model's ability in delineating drought patterns; and 4) analyzing model uncertainty by examining the impact of refined explanatory variables and a reduction in training samples.

## 2 Study area and materials

### 2.1 Study area

The study area is focused on the Three North Protection Forest Project Area, referred to as the Three Norths region. It is located between 73°26′ ~ 127°50′ E and 33°30′ ~ 50°12′ N. The Three North region refers to the part of the area north of Kunlun-Qinling-Daba Mountains, spanning three major geographic areas in northwest, north and northeast China. The region is relatively arid, with precipitation decreasing in a north-to-south and east-to-west pattern across the region, with an average annual precipitation of about 20 ~ 450 mm. Due to the influence of precipitation and other factors, the natural vegetation types are desert, grassland, forest grassland and forest, from west to east. The study area covers diverse climatic zones, featuring a temperate monsoon climate in the northeast and northern regions, while the northwest part exhibits

temperate continental and alpine climates (Fig. S1). The study area exhibits a diverse range of land cover patterns types (Li et al., 2023), as shown in Fig. 1.

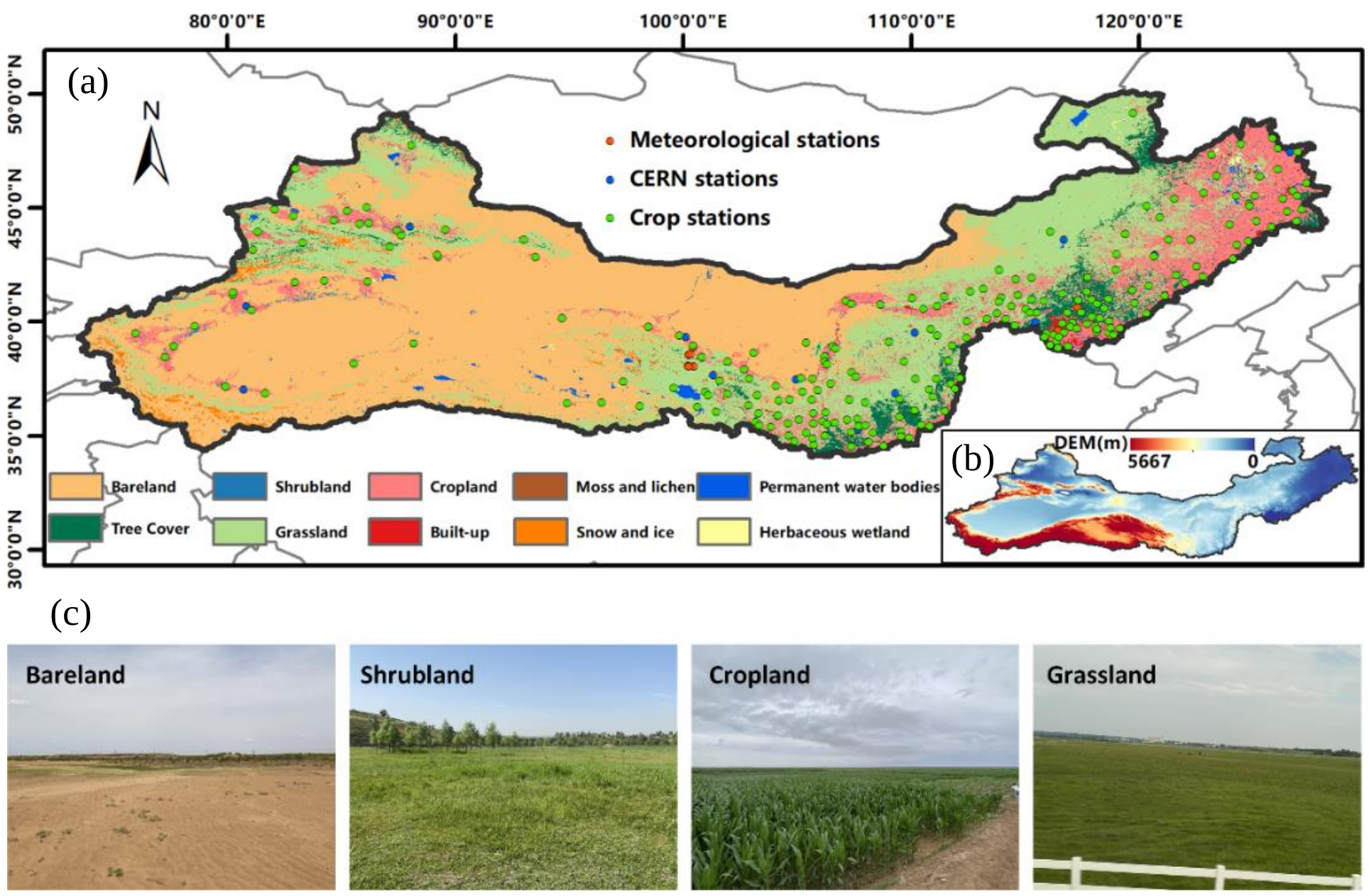

**Figure 1: Geographic Context of the Study Area. (a) Spatial distribution of land types and (b) elevation within the study area. The dots on the maps represent the precise locations of the selected ground stations employed in this research. (c) Photographic representation showcasing characteristic land types in northern China.**

### 2.2 Dataset

The downscaling of satellite-based soil moisture data requires supplementary high-resolution knowledge. Environmental data featuring high spatial resolution, involving land surface temperature (LST), normalized difference vegetation index (NDVI), albedo, elevation, and precipitation, have been identified (feasibility of chosen explanatory factors can be seen in section 3.1). Validation includes a diverse array of ground station datasets, supplemented using drought index data for drought pattern identification. Comparisons are further conducted with outputs from the European Centre for Medium-Range Weather Forecasts (ECMWF) fifth-generation global atmospheric reanalysis (ERA5) and the Noah land surface model. The specific descriptions of dataset used in this study are listed in Table 1.

**Table 1. Summary of available dataset**

| ID | Datasets | Description | Spatial/Temporal resolution |
|---|---|---|---|
| 1 | ESA CCI | Soil Moisture (SM) | 25km/daily |
| 2 | MOD11A1 | Land surface temperature (LST) | 1km/daily |
| 3 | MOD13A2 | Normalized difference vegetation index (NDVI) | 1km/16d |
| 4 | MCD43C3 | Surface albedo (ALB) | 0.05°/daily |
| 5 | SRTM DEM | Elevation | 90m/- |
| 6 | GPM | Precipitation | 0.1°/daily |
| 7 | PDSI | Palmer Drought Severity Index drought index | 4 km/month |
| 8 | ERA5 | Soil moisture (SM) | 0.1°/daily |
| 9 | Noah | Soil moisture (SM) | ~6 km/daily |
| 10 | China Crop Growth and Development and Farmland Soil Moisture Decadal Data Set | In situ soil moisture | -/ten days |
| 11 | Chinese Ecosystem Research Network field stations | In situ soil moisture | -/five days |
| 12 | National Qinghai-Tibet Plateau Scientific Data Center | In situ soil moisture | -/daily |

### 2.2.1 ESA CCI Soil Moisture Data

This study relies on ESA CCI soil moisture data as its primary source. Part of the European Space Agency's Global Observing Program, the CCI Soil Moisture Project aims to create a consistent global soil moisture dataset by integrating data from active and passive microwave sensors. The ESA CCI SM product combines observations from both sensor types, generating active (ESA CCI SM A), passive (ESA CCI SM P), and combined active-passive (ESA CCI SM C) soil moisture products. All ESA CCI SM products offer daily global coverage at a spatial resolution of 0.25 degrees. We use the combined active-passive ESA CCI products from 2003 to 2010, obtained from the ESA data archive (https://www.esa-soilmoisture-cci.org/).

### 2.2.2 Moderate Resolution Imaging Spectroradiometer (MODIS) dataset

The integration of MODIS products within satellite-derived soil moisture downscaling has been extensively employed. The MODIS instrument operates on the Terra and Aqua spacecraft, which covers the largest number of spectral bands of any medium-resolution imager globally and contributes to a range of land and water applications. Specifically, MODIS daily product MOD11A1 is collected to derive LST, while 16-day vegetation index product MOD13A2 is utilized to compute NDVI, both with a spatial resolution of 1 km. Surface albedo is extracted from the daily albedo product MCD43C3. These datasets are procured from the Land Processes Distributed Active Archive Center (LP DAAC) (https://lpdaac.usgs.gov/).

### 2.2.3 Digital elevation model (DEM)

Elevation is a critical topographic attribute in various studies concerning soil moisture downscaling (Mascaro et al., 2011; Wilson et al., 2005). We incorporate the DEM as supplementary data to refine the resolution of ESA CCI soil moisture data. The DEM data are sourced from NASA's Shuttle Radar Topography Mission (SRTM), featuring a spatial resolution of 90 meters.

### 2.2.4 Global Precipitation Measurement (GPM) precipitation

Numerous global and regional studies have verified the relationship between the dynamic patterns of precipitation and the fluctuations in soil moisture (Jones and Brunsell, 2009; Seneviratne et al., 2010). The GPM system, jointly developed by NASA and the Japan Aerospace Exploration Agency (JAXA) as a successor to the Tropical Rainfall Measurement Mission (TRMM), opens a new era in satellite-based global precipitation assessment. Featuring a spatial resolution of 0.1° (approximately 10 km) and a temporal resolution of 1 hour, GPM offers robust spatial representation and timely data. GPM data are sourced from the NASA open data repository (https://pmm.nasa.qov/data-access).

### 2.2.5 In situ measurements

The in situ measurements for model assessment includes ground stations derived from various sources, i.e., the China Crop Growth and Development and Farmland Soil Moisture Decadal Data Set (hereinafter referred to as NZW) (Wang et al., 2016), the in situ measurements obtained from Chinese Ecosystem Research Network field stations (hereinafter referred to as CERN) (Meng et al., 2021), and data from the National Qinghai-Tibet Plateau Scientific Data Center (hereinafter referred to as QXZ) (Gan et al., 2019).

The NZW dataset is compiled from agricultural meteorological decadal reports spanning back to 1991 and acquired from the China Meteorological Science Data Sharing Network (http://cdc.cma.gov.cn/). It features a temporal resolution of ten days, encompassing a substantial time span and wide spatial coverage with a total of 778 stations across the nation. This study merely focusses on the first observation layer (10 cm), which comprises data from 234 stations (Table S1). The soil moisture values are expressed in terms of relative humidity. To facilitate validation, a conversion from relative humidity to volumetric water content is conducted for observed soil moisture.

The CERN dataset comprises 34 stations, covering a period of approximately five days from 2005 to 2014. In our research, we focused on data from 12 stations located within our specific study area (Table 2). We obtained access to this dataset through the National Center for Ecological Science Data (www.cnern.org.cn/data/).

The QXZ dataset is collected from http://data.tpdc.ac.cn. The 5-10 cm in situ soil moisture from six stations are collected, including Arou, Daxing, Guantan, Maliantan, Miyun and Yingke. The daily temporal resolution of these data fulfills the requisite criteria for validation time. Detailed station information is listed in Table 3.

**Table 2. Information of CERN stations**

| ID | Stations | Land Cover | Elevation (m) | Longitude | Latitude |
|---|---|---|---|---|---|
| 1 | Aksu | Farmland | 1028 | 80.85E | 40.67N |
| 2 | Ansai | Farmland | 1083 | 109.31E | 36.85N |
| 3 | Beijing Forest | Forest | 1248 | 115.43E | 39.97N |
| 4 | Qira | Desert | 1306 | 80.70E | 37.01N |
| 5 | Changwu | Farmland | 1200 | 107.67E | 35.25N |
| 6 | Ordos | Desert | 1270 | 110.18E | 39.50N |
| 7 | Fukang | Desert | 460 | 88.00E | 44.15N |
| 8 | Haibei | Farmland | 3280 | 101.33E | 37.66N |
| 9 | Hailun | Farmland | 236 | 126.63E | 47.43N |
| 10 | Linze | Farmland | 1375 | 100.12E | 39.33N |
| 11 | Naiman | Desert | 363 | 120.70E | 42.92N |
| 12 | Shapotou | Farmland | 1350 | 104.95E | 37.45N |

**Table 3.** Information of QXZ stations

| ID | Stations | Land Cover | Elevation (m) | Longitude | Latitude |
|---|---|---|---|---|---|
| 1 | Arou | Grassland | 2295 m | 100.46E | 38.04N |
| 2 | Daxing | Farmland | 20 m | 116.42E | 39.62N |
| 3 | Guantan | Forest | 2835 m | 100.25E | 38.53N |
| 4 | Maliantan | Grassland | 2817 m | 100.30E | 38.55N |
| 5 | Miyun | Farmland | 350 m | 117.32E | 40.63N |
| 6 | Yingke | Farmland | 1519 m | 100.42E | 38.85N |

**2.2.6 Drought index and numerical model outputs**

The Palmer Drought Severity Index (PDSI) (Palmer, 1965) is employed to delineate the critical arid regions for drought assessment. Such an index considers soil moisture and evapotranspiration, and its physical meaning is clear compared to other drought indices. The PDSI dataset is obtained from the Terraclimate dataset (Abatzoglou et al., 2018), with the spatial resolution of ~4 km.

ERA5 reanalysis dataset and Noah land surface model outputs, which are commonly used internationally, are further collected to validate our model results. ERA5, generated from the terrestrial component of the ECMWF ERA5 climate reanalysis (Zhang et al., 2021b), offers an hourly temporal resolution and an approximate spatial resolution of 10 km. This dataset delivers soil moisture at various depths from 1981 to the present. Here, we select the data from the topmost layer (0-7 cm). Another modeled surface (0-10 cm) soil moisture is derived from Noah model. These simulations include a spatial resolution of approximately 6 km, effectively covering the extensive expanse of the Chinese Loess Plateau region. The detailed execution of the Noah model will be elaborated upon in Section 3.5.

## 3 Methods

The framework primarily involves Bayesian model averaging (BMA) along with four regression approaches: random forest, multiple linear regression (MLR), support vector machine, and extreme gradient boosting (XG). It entails several critical step: i) assessing the feasibility of chosen explanatory factors from diverse knowledge sources through importance and correlation analysis, subsequently partitioning the region based on climate and hydrology patterns; ii) training and applying multiple machine learning techniques to downscale coarse resolution soil moisture, relying on an adaptable spatial and temporal window searching strategy; iii) leveraging BMA approach to merge the outputs of four regression methods, based on the suitable partitioned regional zoning; and iv) evaluating the downscaled results by comparing with a range of in-situ measurements and model simulations. Fig. 2 shows the flow chart of the study.

### 3.1 Feasibility of chosen explanatory factors

Albedo, NDVI, precipitation, LST, and DEM are identified as input explanatory variables. These variables can be collected through reliable and readily available data at large scales, thereby enhancing the model's potential for extension to diverse regions. These variables have been consistently associated with soil moisture in previous downscaling studies (Liu et al., 2023; Ranney et al., 2015; Song et al., 2014; Zhang et al., 2022). Specifically, atmospheric variables (e.g., LST, albedo, and precipitation) maintain temporal variability, while geophysical variables (e.g., DEM) capture spatial variability of the downscaled soil moisture. Furthermore, the incorporation of NDVI aims to characterize the influence of vegetation on the spatiotemporal dynamics of soil moisture (Abbaszadeh et al., 2019).

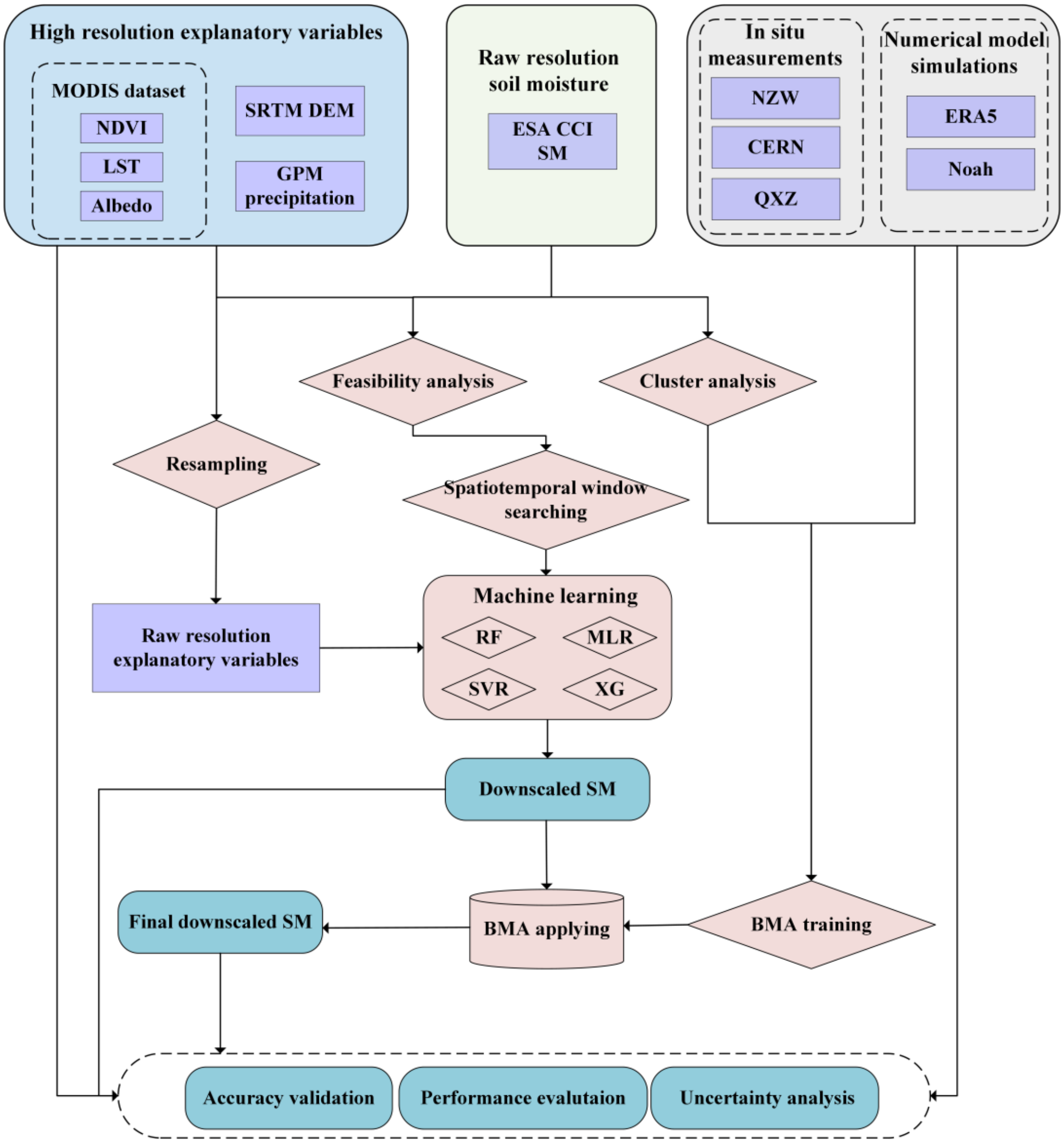

**Figure 2: Flowchart of the presented framework**

To validate the feasibility of the selected explanatory variables, we compute the importance percentage and SHAP (Shapley
200 Additive exPlanations) values of the five auxiliary datasets with respect to ESA CCI soil moisture from 2009 to 2010. The utilization of the random forest model enables the quantification of the explanatory variables' impact probabilities, with higher values indicating a stronger ability to characterize soil moisture. As depicted in Fig. 3(a), all five variables exhibit an importance exceeding 0.3. The highest significance is attributed to DEM with a value of approximately 0.69, closely trailed by LST, precipitation, and albedo, all surpassing 0.4. The lowest percentage is attributed to NDVI. Results substantiate the
205 high relevance of these five auxiliary datasets to ESA CCI soil moisture data, validating the reasonable selection of these

explanatory variables. Moreover, the robustness of these variables is reinforced by their high SHAP values and importance scores obtained from other models (Fig. S2).

We further examine the relationships between utilized variables and ESA CCI soil moisture through linear regression analysis, and Fig. 3(b) delineates a robust correlation between them. NDVI, precipitation, and DEM exhibit a positive correlation with soil moisture, whereas albedo and LST display a negative correlation. The most pronounced correlation emerges with LST at approximately -0.4, attributable to the essential role of LST in surface energy flux regulation and distribution (Im et al., 2016). Succeeding in magnitude are NDVI and albedo, with 0.35 and -0.28 correlations, respectively, underscoring NDVI's capacity to represent vegetation coverage (Piles et al., 2011). Collectively, these five explanatory variables exhibit strong correlations with ESA CCI soil moisture, thus affirming their feasibility for soil moisture downscaling.

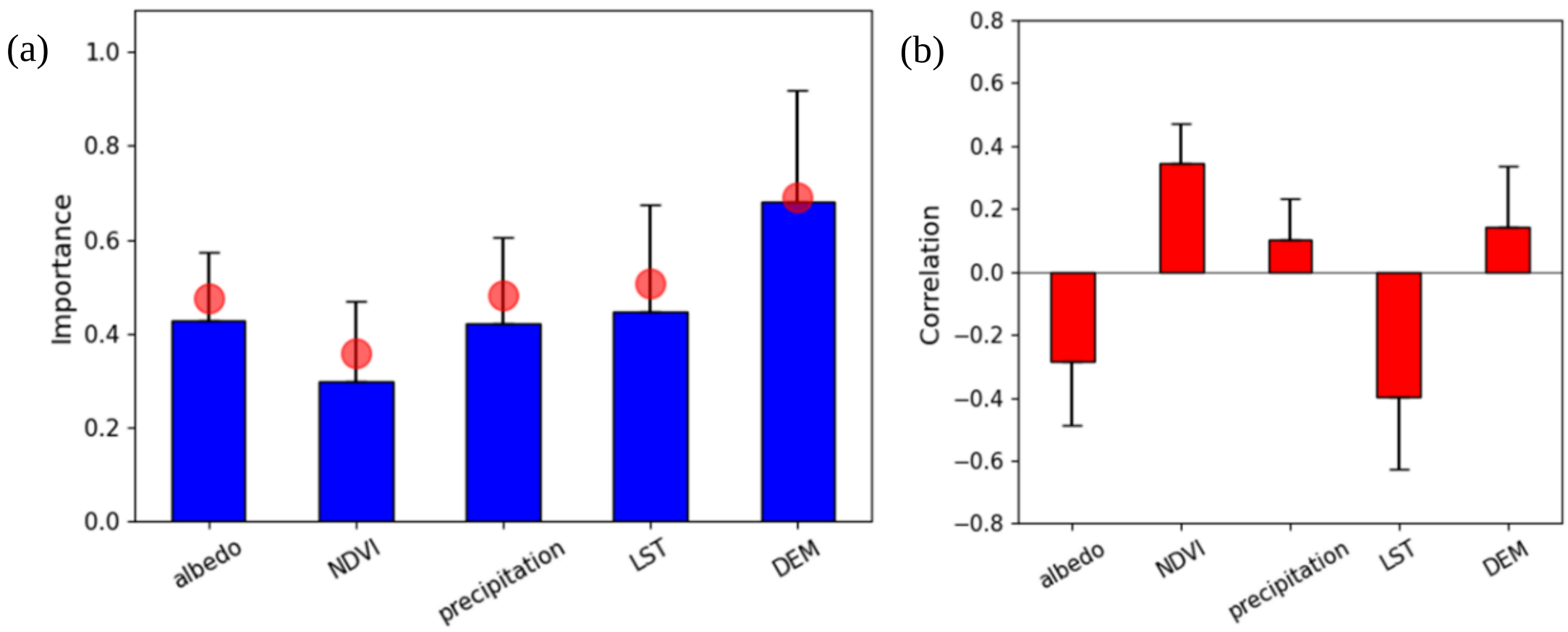

**Figure 3: Assessment of explanatory variables' feasibility. (a) Average (blue bar) and standard deviation (error bar) of permutation-based importance of explanatory variables concerning soil moisture, with corresponding SHAP values depicted as red circles. (b) Average (red bar) and standard deviation (error bar) of the Pearson correlation coefficient denoting explanatory variables' association with soil moisture.**

### 3.2 Machine learning methods

The fundamental principle beneath the soil moisture downscaling model relies on the establishment of a robust nonlinear functional relationship connecting a multitude of diverse input parameters to sequential soil moisture. This concept can be expressed as follows:

$$SM = f(X) + \varepsilon , \quad (1)$$

In this equation, the variable SM on the left-hand side symbolizes soil moisture, while the vector $X = \{x_1, x_2, \ldots, x_k\}$ encompasses an array of input parameters, i.e., albedo, NDVI, precipitation, LST, and DEM. The core of this equation resides in the nonlinear function f, which captures the dependencies between these input parameters and the dynamics of soil moisture.

### 3.2.1 Random forest

The random forest approach, a modified version of the decision tree model, utilizes an ensemble of multiple regression trees to reach decisions (Long et al., 2019). This model has gained prominence in soil moisture data analysis compared to other machine learning methods (Liu et al., 2018).

During the training phase, RF constructs multiple decision trees and then averages the predictive outcomes of these trees to generate the final output. The RF outcome is derived from the average of predictions from each individual decision tree, as demonstrated by:

$$f(SM|X) = \frac{1}{n}\sum_{i=1}^{n} f_i(SM|X) , \quad (2)$$

where n represents the count of regression trees, $f(SM|X)$ signifies the integrated decision tree, and $f_i(SM|X)$ denotes the sub-decision tree derived from the initial SM within the input parameters X.

### 3.2.2 Multiple linear regression

The principle of MLR (Wilks, 2011) is relatively simple, mainly using statistical methods to establish a linear relationship between soil moisture and several explanatory factors as follows:

$$SM = a_0 + a_1x_1 + a_2x_2 + \cdots + a_{k-1}x_{k-1} + a_kx_k , \quad (3)$$

where $\{x_1, x_2, \ldots, x_k\}$ is the auxiliary variable; $\{a_1, a_2, \ldots, a_k\}$ is the partial regression coefficient corresponding to each auxiliary data; k is the number of auxiliary variables.

### 3.2.3 Support vector regression

SVR is a machine learning method rooted in the utilization of nonlinear transformations of covariate, pioneered by Vapnik (Vapnik, 1999). This modeling approach, characterized by its ability to grasp the nonlinear relationships, embarks upon a training phase wherein input datasets and corresponding target outputs undergo training (Kim et al., 2018). During this iterative process, the model strikes a harmonious balance between its output predictions and the actual veritable values by assimilating input training samples, a quintessential hallmark of SVR's quest for fidelity. SVR can be described as a mapping relationship in the nonlinear space represented as follows:

$$SM = f(x_1, x_2, \ldots, x_k) = \sum_{i=I}^{m} \omega_i x_i + b , \quad (4)$$

where $x_i$ denotes the value of each dimension in the training set; i denotes the variable of each dimension; ω denotes the variable coefficient; and b denotes the bias.

### 3.2.4 Extreme gradient boosting

The fundamental principle of the XG algorithm revolves around training weak classifiers that are progressively integrated based on negative gradient insights derived from the existing model's loss function. These trained weak classifiers are incorporated into the existing model in an accumulative manner. This technique represents an enhanced form of gradient

boosting, generating a sequence of sequential regression trees. Each of these trees fits the residuals of the previous one to a target value until a predefined depth or convergence criterion is met (Chen and Guestrin, 2016; Zhang et al., 2021a). The objective function of XG can be mathematically represented as follows:

$$\mathrm{Obj}^{(t)} = \sum_{i=1}^{n}[\mathrm{l}(\mathrm{y_i}, \widehat{\mathrm{y_i^{t-1}}}) + \mathrm{g_i f_t(x_i)} + \frac{1}{2}h_\mathrm{i}\mathrm{f_t^2(x_i)}] + \sum_{i=1}^{t}\Omega(\mathrm{f_i}) \, , \tag{5}$$

where $\mathrm{y_i}$ denotes the actual value of the ith sample; $\widehat{\mathrm{y_i^{t-1}}}$ represents the predicted value of the ith sample given at the t-1 step model; $\mathrm{l}(\mathrm{y_i}, \widehat{\mathrm{y_i^{t-1}}})$ constitutes the loss function derived formed the predicted and true values; $\mathrm{g_i}$ signifies the first-order derivative and $\mathrm{h_i}$ corresponds to the second-order derivative of the loss function; $\mathrm{f_t}$ denotes the base model; and $\Omega$ signifies the canonical term, influenced by the count of leaf nodes in the decision trees and the application of regularization to the weights of these leaf nodes.

### 3.2.5 Model parameters selection

The model's critical hyperparameters are optimized to enhance accuracy and prevent overfitting. The explanatory variable dataset is subsampled to 0.25° to match the resolution of the raw soil moisture through cubic convolution resampling techniques. This resulting dataset is referred to as the original training dataset of the model. Each machine learning approach in this study is then applied to the training dataset, with parameters optimized for minimal Root Mean Square Error (RMSE) via a 10-fold cross-validation process. An integral aspect of our methodological framework involves randomly allocating nine-tenths of the spatial sample for model fitting, reserving the remaining one-tenth for validation. Parameter value ranges for each hyperparameter are chosen based on a comprehensive review of relevant scientific literature and best practices in the machine learning community (Table S2).

### 3.3 Bayesian model averaging

This study employs Bayesian model averaging (Raftery et al., 2005) to create an ensemble of downscaled results from various machine learning methods. BMA is a statistical approach rooted in Bayesian theory, designed to incorporate the inherent uncertainty of models during data processing. Within this framework, the method assigns appropriate weights to multiple downscaled models based on the posterior probability of each model's predictive accuracy and alignment with prior knowledge. This approach addresses the challenges of uncertainty and singularities often associated with individual models. Particularly, the incorporation of prior knowledge from point-wise data (i.e., in-situ ground measurements) facilities mitigate the impact of outliers or noisy data points on the ensemble's predictions.

Assume that y is the combined predictor variable, i.e., the downscaled soil moisture from individual machine learning.$\mathrm{D} = \{\mathrm{y_1, y_2, \ldots, y_T}\}$ is the measured samples, i.e., primary in situ measurements in 2003-2008. $\mathrm{M} = \{\mathrm{M_1, M_2, \ldots, M_k}\}$ is the model space composed of all possible prediction models. According to the full probability law, the expression of the predicted Probability Density Function (PDF) of the combined prediction y of the BMA method:

$$p(y|D) = \sum_{i=1}^{k} p(M_i|D)\, p(y|M_i, D) \, , \tag{6}$$

where $p(y|M_i, D)$ is the prediction PDF based on model $M_i$; $p(M_i|D)$ is the posterior probability that model $M_i$ is correctly predicted with measured samples D, reflecting the degree of fit of model $M_i$ to the measured samples. The sum of the posterior model probabilities is equal to 1, i.e., $\sum_{i=1}^{k} p(M_i|D) = 1$. These probabilities can be interpreted as weighting factors. The prediction results of BMA can be expressed as:

$$E(y|D) = \sum_{i=1}^{k} p(M_i|D)\, E[p(y|M_i, D)] = \sum_{i=1}^{k} w_i f_i \,, \tag{7}$$

where $w_i$, $f_i$ represent the weight and prediction result of $M_i$, respectively.

In theory, calculating $p(M_i|D)$ of a model involves computing the likelihood function for each model, multiplying it by the prior probability of each model, and dividing by the marginal likelihood. However, this method is rarely employed in practice due to the complexity of computing the likelihood function and prior distribution, especially for complex models with high-dimensional parameter spaces. Instead, iterative estimation techniques such as Markov Chain Monte Carlo methods are commonly used. In our study, we utilized Markov Chain Monte Carlo Cube (MC3) for this purpose.

### 3.4 Cluster analysis and spatiotemporal window searching

Considering the potential variability in soil moisture-explanatory factor relationships across our vast spatial domain, the uncertainty in the global regression model may increase, particularly with limited training samples. Uneven distribution of ground training data in vast spatial domains could lead to fewer data points in certain regions, posing a challenge to the model's ability to capture fine characteristics and reducing overall accuracy. By treating the entire area as homogeneous, crucial geographical and climatic details are overlooked, exacerbating uncertainty. To mitigate this, we employ cluster-based models to better capture spatiotemporal soil moisture variation compared to a global model. A spatial partitioning of the study area is implemented before applying machine learning models, as depicted in Fig. 4(a). Previous research has employed cluster analysis to segment study areas grounded in distinctive characteristics or data attributes (Xiao et al., 2018b). Here we calculate the multi-year average of ESA CCI soil moisture data spanning 2003 to 2010. This multi-year average dataset underwent k-means clustering analysis to yield a division scheme of the study area. Subsequent to this partitioning, the BMA model is trained utilizing the outcomes of this segmenting process. The BMA model's output is aligned with the trained model specific to each partition, harmonizing the partitioning strategy's application throughout the model training and result output stages.

Efficiently exploring informative covariates is a challenge in machine learning models. Here, we adopt a spatiotemporal approach (Liu et al., 2020) to effectively capture the spatial and temporal fluctuations in soil moisture and its correlated covariates. This approach centres on regional subsets, aiming to involve a greater array of relevant pixels in the regression process (Fig. 4(b)). An adaptable mechanism is used to determine the optimal spatiotemporal window size, relying on two critical parameters: the spatial window size and the temporal duration in terms of days. This strategy involves incrementally increasing these two parameters from initial values until a specific criterion is met, which mandates that the available pixels within the window encompass at least eight times the included explanatory variables. The initial values are set at a spatial

window size of 3 and a duration of days of 1. By introducing upper limits to these parameters, the process is ensured to terminate, accounting for potential gaps in the satellite dataset. Through a sensitivity analysis conducted with an independent dataset, maximum values for these parameters are chosen for the period spanning 2003 to 2010. Testing various combinations of parameters involved exploring different temporal durations from 1 to 5 days, increasing by 1 day increments, and spatial window sizes from 3 to 10, with 1 unit increments. This yields the optimal parameter values are a temporal duration of 3 days and a spatial window size of 5. The sensitivity analysis reveals consistent optimal parameter values across the seven climate regions, likely due to the focus on model structure rather than sample characteristics.

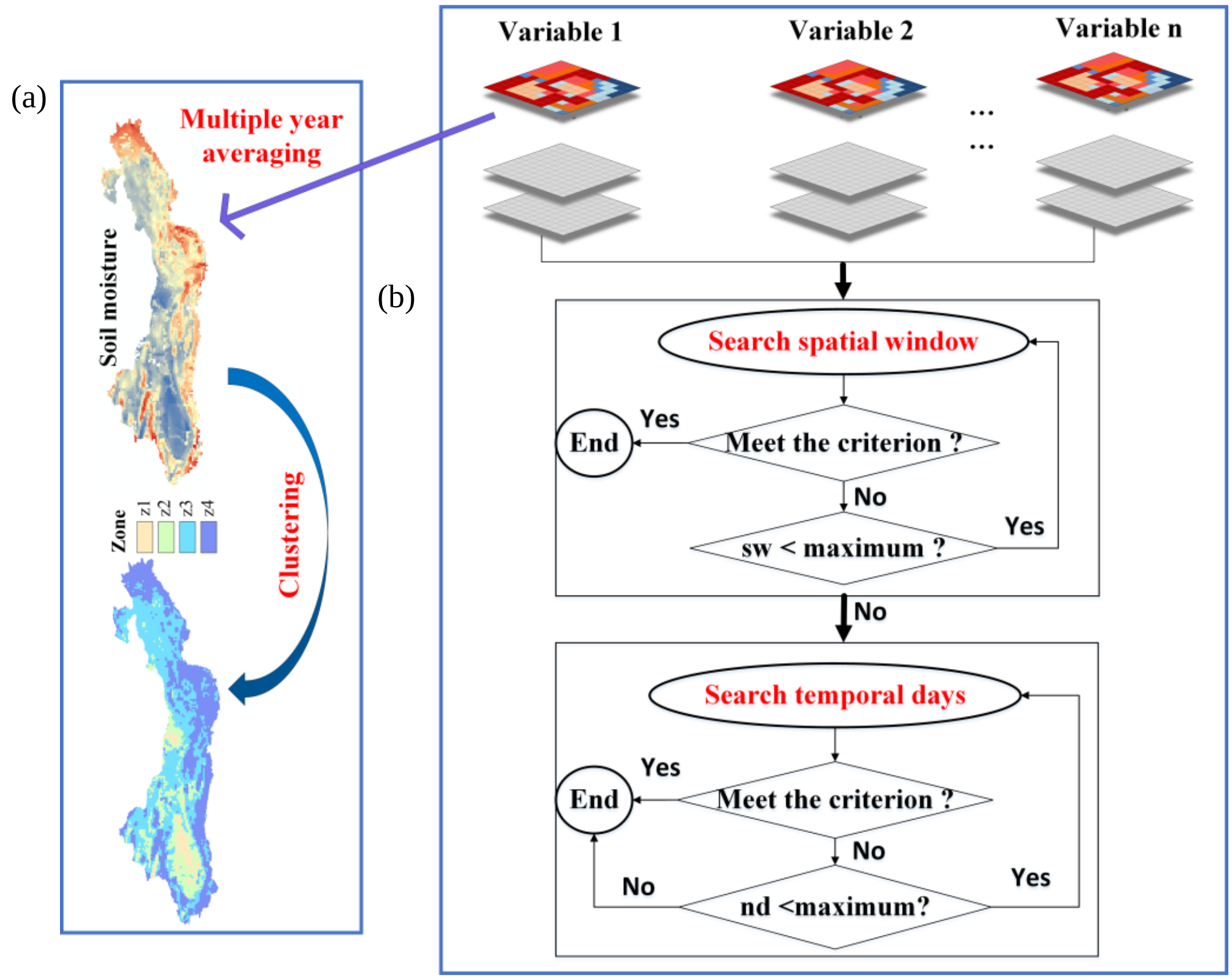

**Figure 4: Strategies for machine learning optimization. (a) Schematic illustrating regional clustering applied to the BMA model. (b) Schematic outlining the strategy for determining spatiotemporal windows in machine learning regression which involves defining two key parameters: the spatial window size (sw) and the temporal duration in terms of days (nd).**

### 3.5 Evaluation analysis from diverse perspectives

Evaluating proposed model from a multitude of perspectives is of importance in maintaining the accuracy and hydrological responsiveness of satellite-derived soil moisture. We initially validated downscaled soil moisture through comparison with in situ measurements from three ground station networks. A comparative analysis is further conducted across various machine learning models. Specifically, the accuracy validated is quantitatively performed using statistical metrics including the

correlation coefficient (R), mean absolute error (MAE), RMSE, and unbiased root mean squared error (ubRMSE). The formulas for these calculations are as follows:

$$R = \frac{\sum(X_i-\bar{X})(Y_i-\bar{Y})}{\sqrt{\sum(X_i-\bar{X})^2(Y_i-\bar{Y})^2}}, \quad (8)$$

$$RMSE = \sqrt{\frac{\sum_{i=1}^{n}(X_i-Y_i)^2}{n}}, \quad (9)$$

$$MAE = \frac{1}{n}\sum_{i=1}^{n}|X_i - Y_i|, \quad (10)$$

$$ubRMSE = \sqrt{RMSE^2 - BIAS^2}, \quad (11)$$

where n represents the count of observations, and $X_i$ and $Y_i$ are the in situ and the modeled soil moisture, respectively.

We employ surface soil moisture from the ERA5 and Noah to conduct cross validation, aiming to discern the congruence between the results of our model and internationally recognized datasets. We specifically run the Noah model at a spatial resolution of 0.05° (~6 km). The meteorological input data are sourced from the China meteorological forcing dataset, with a 3-hour temporal resolution and 1° spatial resolution. This dataset comprises seven essential parameters, encompassing 2-meter air temperature, 10-meter wind speed, specific humidity, air pressure, downward shortwave, longwave radiation and accumulated precipitation. Recognizing the possibility of the Noah model's underestimation of peak flows for thermal and energy fluxes, we opt to utilize surface dynamic variability as a replacement for the traditional static surface parameters. This subset of parameters, comprising leaf area index, surface albedo, and green vegetation fraction, is updated through seasonally varying satellite observations from MODIS. To ensure precise allocation of distinct vegetation and soil types, the integration of land cover and soil texture maps from the State Soil Geographic Database/Food and Agriculture Organization is employed. A more extensive elaboration can be referenced in our previous research endeavours (Liu et al., 2022b; Liu et al., 2023; Liu et al., 2020).

A drought assessment is further performed by focusing on drought-prone regions of interest. Within these regions, a temporal analysis is applied to ground station measurements, CCI soil moisture, and the results obtained from the machine learning downscaling method.

Uncertainty analyses are conducted to evaluate the model's robustness. A leave-one-out parameter analysis is performed to assess the influence of the explanatory factors on the model outcomes. To explore the impacts of training samples on the mode accuracy, we iteratively eliminate subsets of training samples from the Bayesian model and subsequently assess the implications on the validation outcomes.

## 4 Results and discussion

### 4.1 Spatial pattern of downscaled soil moisture

Utilizing a machine learning downscaling framework, the daily ESA CCI soil moisture with an original spatial resolution of approximately 25 km is downscaled to a finer resolution of 1 km. Fig. 5 (and Fig. S3) presents the spatial patterns of both the

raw ESA CCI soil moisture and the downscale soil moisture of four distinct machine learning methods, along the results obtained through BMA ensemble model for the two consecutive years of 2009 and 2010. Specifically, the months ranging from April to October of both 2009 and 2010 are considered for visualization.

The results from individual machine learning methodology align with the patterns observed in the original ESA CCI. This congruence is pronounced within areas exhibiting both high and low soil moisture values. The results illustrate high soil moisture levels in the southern and northeastern parts of the study region, in contrast to comparatively lower soil moisture content observed in the central and western regions. This spatial variability can be explained by underlying factors such as land use composition and precipitation dynamics (Li et al., 2011). The arid and desert-like land use in the western portion of the study area, combined with limited precipitation, contributes to the lower soil moisture observed.

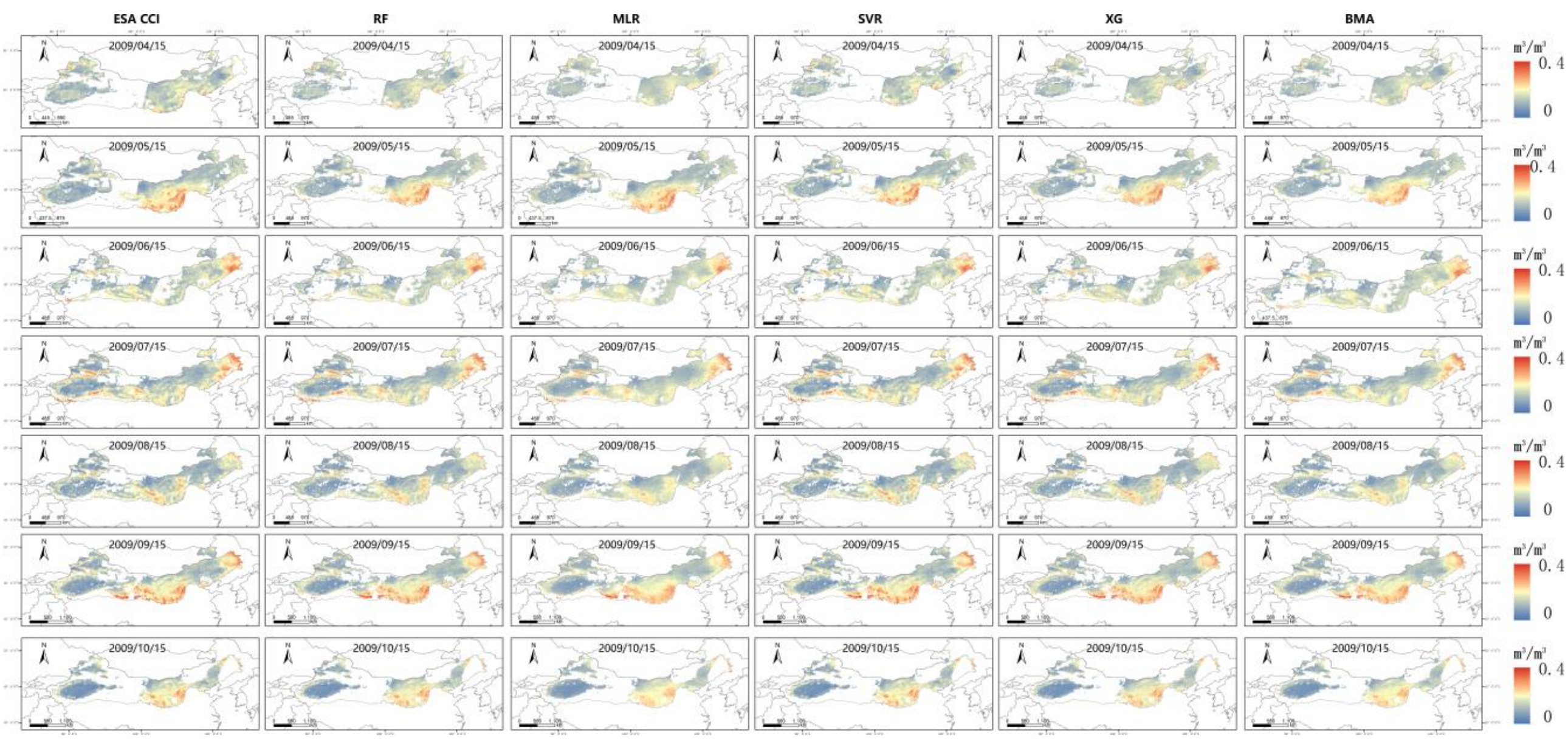

**Figure 5: Spatial distribution of soil moisture across six data sources, representing the 15th day of April-October 2009. Columns, from left to right, show the 25-km ESA CCI soil moisture and the 1-km downscaled soil moisture derived from random forest (RF), multiple linear regression (MLR), support vector regression (SVR), extreme gradient XG Boost (XG), and Bayesian model averaging (BMA), respectively.**

Examining histograms visually enhances the coherence between ESA CCI and downscaled results. In Fig. 6(a) (and Fig. S4), all techniques align with the dataset's broader trajectory, but SVR exhibits the most similarity with ESA CCI soil moisture. The concentration of soil moisture values is notably within the 0.15 to 0.2 $m^3/m^3$ range, highlighting the arid nature of the study area. It is evident that the downscaled data produced by the BMA method exhibit more pronounced differences compared to the original data, particularly in terms of histogram distributions shifting towards the peak. This implies that the downscaled method effectively captures the disparities between the 25km and 1km products. These differences in the

histograms indicate a more concentrated distribution of data, which can mitigate the issue of overestimation and underestimation by machine learning models.

Figure 6(b) shows the box plots encompassing twelve months of ESA CCI data and the corresponding BMA ensemble outcomes. The disparities between the two datasets are relatively inconspicuous across most of the months, with January and December demonstrating lower values overall. Such phenomenon may be related to the inherent nature of ESA CCI soil moisture values during those months (Yuling et al., 2022). While most of the downscaling results exhibit lower values compared to the original ESA CCI values during most months, this variance is not of substantial magnitude. This pattern can be attributed to the inherent characteristics of the BMA ensemble approach, which combines multiple machine learning outcomes to prevent excessively high or low values.

We further check the box plots categorized according to soil dryness and moisture levels (Fig. 6(c)) Both the ESA CCI data and BMA ensemble products demonstrate a discernible upward trend in soil moisture as the soil transitions from a dry to a wet state. This underscores the efficacy of both datasets in capturing the dynamic spectrum of dry and wet soil conditions. Meanwhile, the values obtained from BMA ensemble method exhibit a lower overall profile compared to the ESA CCI values, an alignment reflected in Fig. 6(b). In general, the degree of concordance between the BMA ensemble-derived results and the original ESA CCI data is underscored.

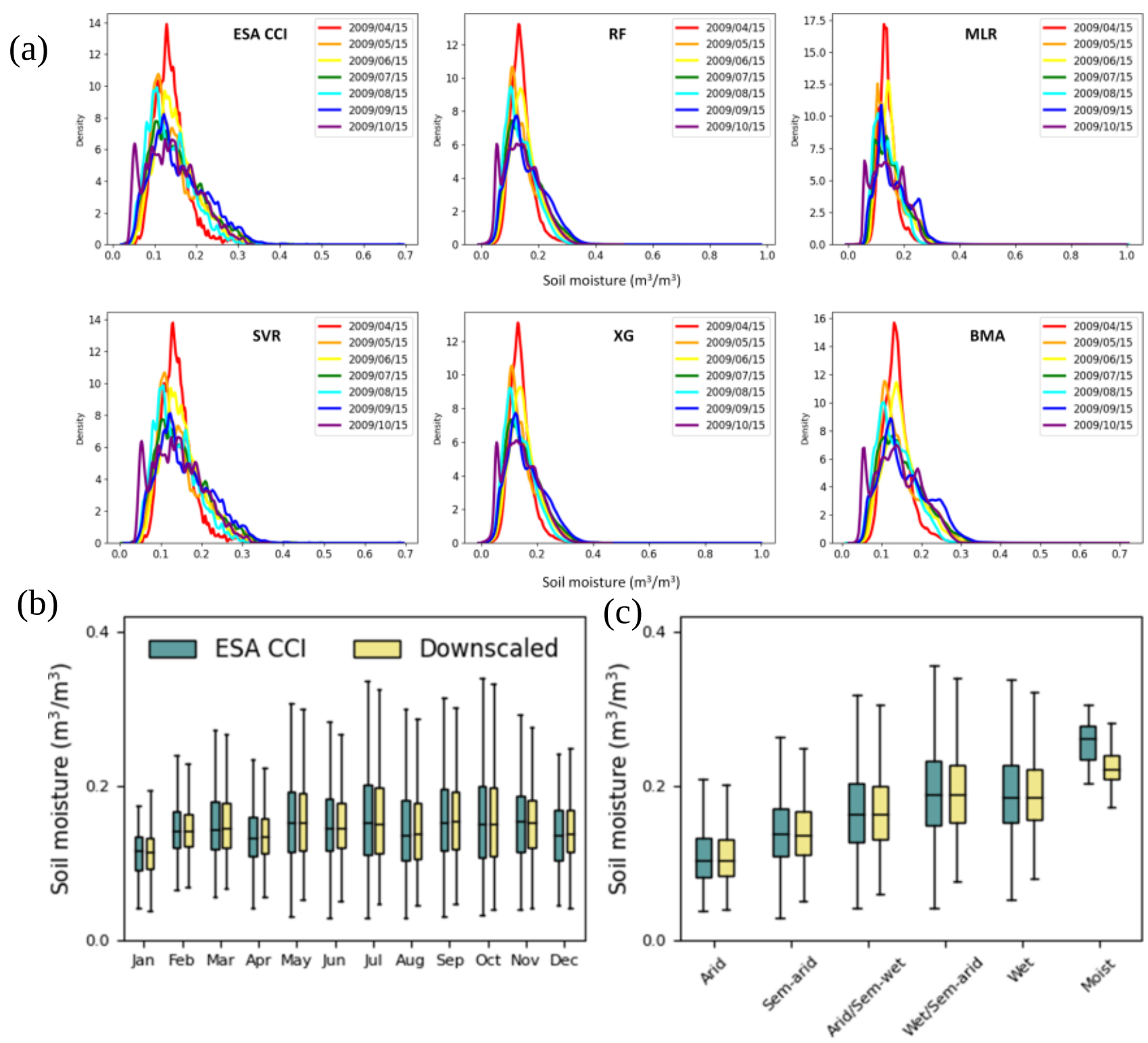

**Figure 6: Soil moisture distribution analysis. (a) Histograms illustrating soil moisture data on the 15th day of April-October 2009. Histograms show the comparison between 25-km ESA CCI soil moisture and 1-km downscaled soil moisture using random forest**

**(RF), multiple linear regression (MLR), support vector regression (SVR), extreme gradient XG Boost (XG), and Bayesian model averaging (BMA). (b) Box plots demonstrating the monthly averages of ESA CCI and BMA ensemble soil moisture. (c) Box plots showing the averaged soil moisture of ESA CCI and BMA ensemble grouped by wet and dry conditions. Medium values are marked by black lines, while boxes and whiskers denote the 25th to 75th percentiles and the 5th to 95th percentiles, respectively.**

### 4.2 Accuracy validation of downscaled soil moisture

The validation of BMA ensemble outcomes is performed against three ground station datasets, as presented in Fig. 7. Excluding a few isolated instances characterized by lower R values and higher MAE, the dominated trend demonstrates a favourable accuracy. For most monitoring stations, higher R values correspond with lower MAE values. The stations characterized by lower R values and higher MAE values are predominantly situated within the western reaches of the study area. This is understandable since the spatial distribution of monitoring stations within the study domain exhibits an uneven dispersion. The absence of in situ data in the western desert-dominated region potentially weak the model training, exerting a negative effect on the resultant model accuracy. Meanwhile, the arid nature of the western region, marked by its desert land use type, results in considerable variability in temperature and energy radiation patterns. Additionally, there are clusters of low-accuracy points along the east boundary, which are known for their diverse and complex land use patterns, including wetland marshes, farmland, and fishing areas. The intricate interplay of these land use types, combined with variations in weather patterns and ecosystem dynamics, poses challenges to the accuracy of our model.

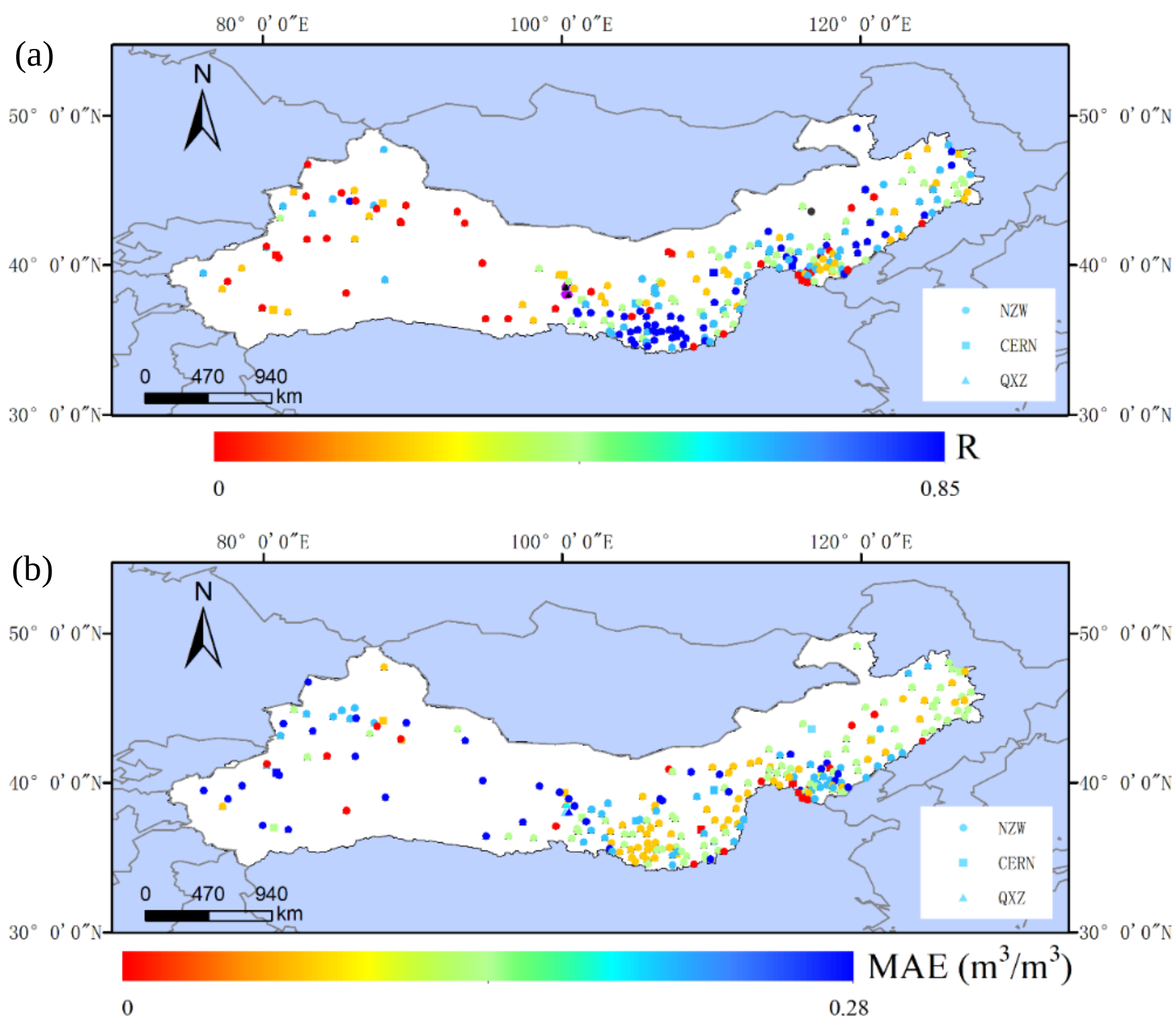

Hydrology and Earth System Sciences Open Access EGU Discussions

**Figure 7: Performance evaluation of downscaled soil moisture. (a) Correlation coefficient (R) and (b) mean absolute error (MAE) illustrating the accuracy of BMA-based downscaled soil moisture when compared against in situ measurements from three ground station networks.**

Figure 8 shows a scatter plot of the correspondence between downscaled soil moisture and ground measurements. It is observed an evident enhancement in accuracy achieved by the BMA ensemble relative to the original ESA CCI data. Regarding NZW network, the ESA CCI data exhibits an R of 0.321, accompanied by corresponding RMSE and ubRMSE values of 0.138 and 0.095 $m^3/m^3$, respectively. In contrast, the BMA ensemble results produce an incremented R value of 0.342, along with reduced RMSE and ubRMSE values of 0.137 and 0.093 $m^3/m^3$, respectively. Similar patterns are observed across CERN network, wherein the BMA outcomes present a modest increase in R by 0.003, along with a corresponding decrease in ubRMSE by 0.001 $m^3/m^3$ in comparison to the ESA CCI data. The QXZ network displays a more pronounced advancement, with the BMA results exhibiting a higher R of 0.035, accompanied by a reduction in ubRMSE by 0.003 $m^3/m^3$relative to the original ESA CCI data.

The soil moisture across distinct temporal periods is also checked, particularly during the monsoon period (May-September). As for the NZW stations, no apparent disparity is discerned between station measurements during monsoon and non-monsoon seasons. Both ESA CCI and BMA results demonstrate consistent patterns, with an overall underestimation of soil moisture levels. The in situ measurements of CERN stations exhibit high values during the monsoon season in contrast to other months. Conversely, the ESA CCI and BMA downscaling outcomes depict lower values during the monsoon period than other months. As for QXZ stations, both ESA CCI and BMA downscaling outcomes consistently underestimated soil moisture across all periods. In general, station measurements during the monsoon season exhibit lower values compared to other months, a trend that is found in the ESA CCI and BMA data. The outcomes from both ESA CCI and BMA data broadly align with the observed trends at the station-specific level. This reinforces the credibility of these results in reflecting variations in soil moisture different temporal times. However, it also underscores the challenges in accurately capturing soil moisture dynamics during the monsoon season, especially in certain large geographic regions (Park et al., 2017a).

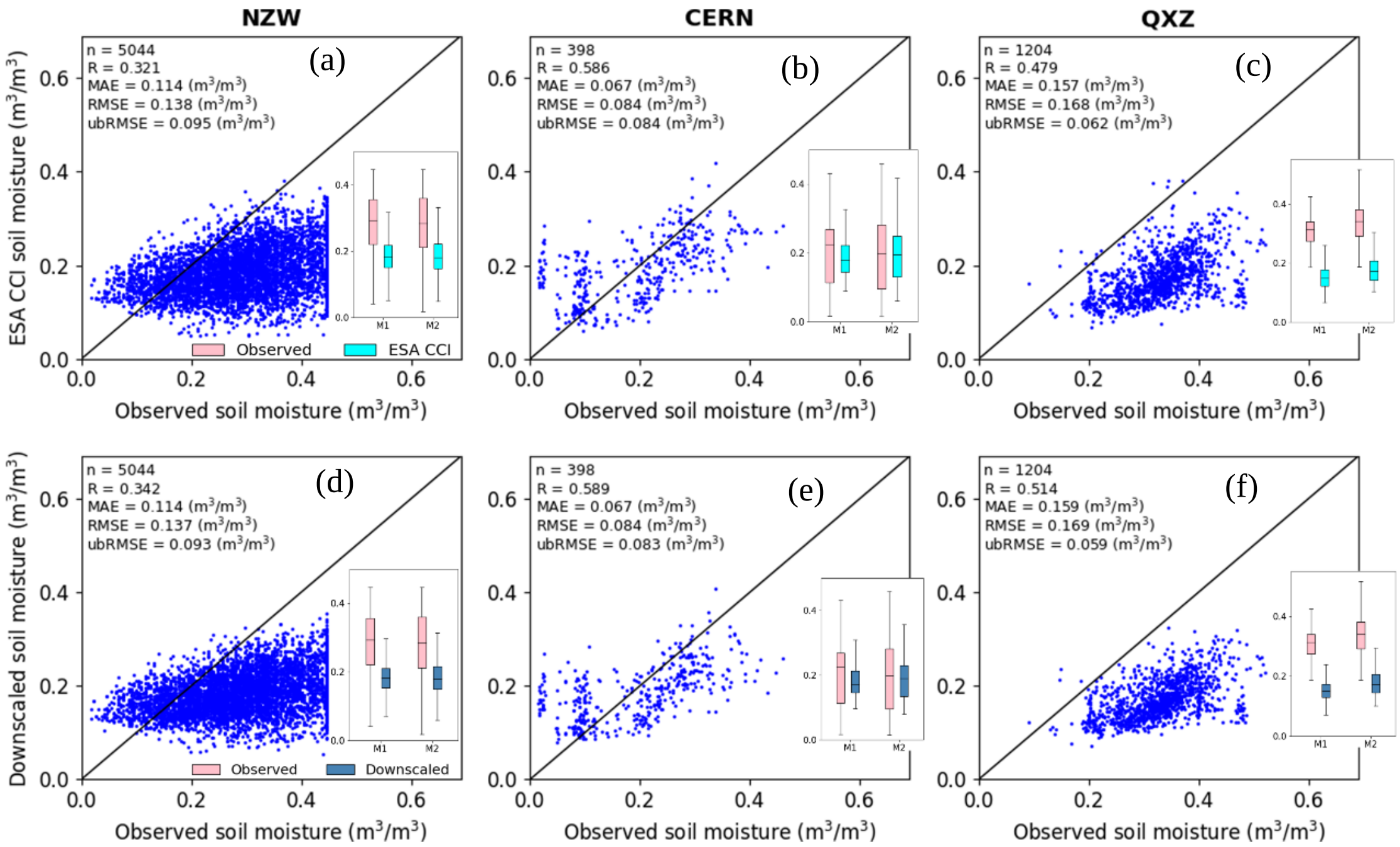

**Figure 8: Accuracy evaluation of soil moisture. (a)-(c) Scatter plots of ESA CCI soil moisture against in situ measures from three ground station network. (d)-(f) Scatter plots of BMA downscaled soil moisture against in situ measures from the same three ground station network. In the subfigures located in the lower corners of the panels, box plots depict the distribution of ESA CCI and BMA downscaled soil moisture categorized into monsoon seasons (May-September, denoted as M1) and non-monsoon seasons (Rest of the year, denoted as M2). The central black lines indicate median values, while the boxes and whiskers represent the interquartile range (25th to 75th percentiles) and the full range (5th to 95th percentiles), respectively.**

### 4.3 Comparison of different machine learning models

The downscaled soil moisture produced by the BMA ensemble model is compared against the results from four individual regression methods (Table 4). The relative efficacy of the four machine learning methods varies across diverse stations, while the BMA results outperform them. The R of the BMA ensemble results against in situ measurements surpasses that of the four individual machine learning methods. Additionally, the RMSE and MAE exhibit lower values in most instances for the BMA results. Regarding NZW stations, MLR exhibits higher accuracy among the four individual machine learning methods. In contrast, SVR shows higher accuracy for CERN stations and shows optimal performance in QXZ stations. The BMA approach demonstrates better overall performance, underscoring its capacity to combines the strengths inherent in each individual method. This aligns with earlier studies that employed BMA approach to harness the complementary attributes of different methodologies in yielding outcomes of enhanced reliability and precision (Miao et al., 2013).

**Table 4.** Comparison of BMA and individual machine learning

| Stations | N | R | | | | |
|---|---|---|---|---|---|---|
| | | RF | MLR | SVR | XG | BMA |
| NZW | 5044 | 0.323 | 0.338 | 0.321 | 0.325 | 0.342 |
| CERN | 263 | 0.567 | 0.617 | 0.629 | 0.573 | 0.642 |
| QXZ | 1204 | 0.477 | 0.480 | 0.478 | 0.468 | 0.514 |
| **Stations** | **N** | **RMSE（$m^3/m^3$)** | | | | |
| | | RF | MLR | SVR | XG | BMA |
| NZW | 5044 | 0.138 | 0.136 | 0.138 | 0.137 | 0.137 |
| CERN | 263 | 0.073 | 0.073 | 0.073 | 0.075 | 0.071 |
| QXZ | 1204 | 0.169 | 0.169 | 0.168 | 0.169 | 0.169 |
| **Stations** | **N** | **MAE（$m^3/m^3$)** | | | | |
| | | RF | MLR | SVR | XG | BMA |
| NZW | 5044 | 0.114 | 0.113 | 0.114 | 0.114 | 0.114 |
| CERN | 263 | 0.060 | 0.061 | 0.061 | 0.062 | 0.058 |
| QXZ | 1204 | 0.157 | 0.158 | 0.156 | 0.158 | 0.158 |

We analysed time series data, incorporating in situ measurements, downscaled soil moisture from four distinct machine learning methods, and the BMA ensemble, comparing them with ESA CCI data. Specifically focusing on two QXZ stations and two NZW stations (Fig. 9), all six datasets exhibit a level of agreement with ground-based soil moisture dynamics. Despite occasional overestimation or underestimation, the main trends consistently align with observed values. This alignment, especially during extreme wet or dry periods, highlights the BMA ensemble's accuracy in capturing anomalous conditions.

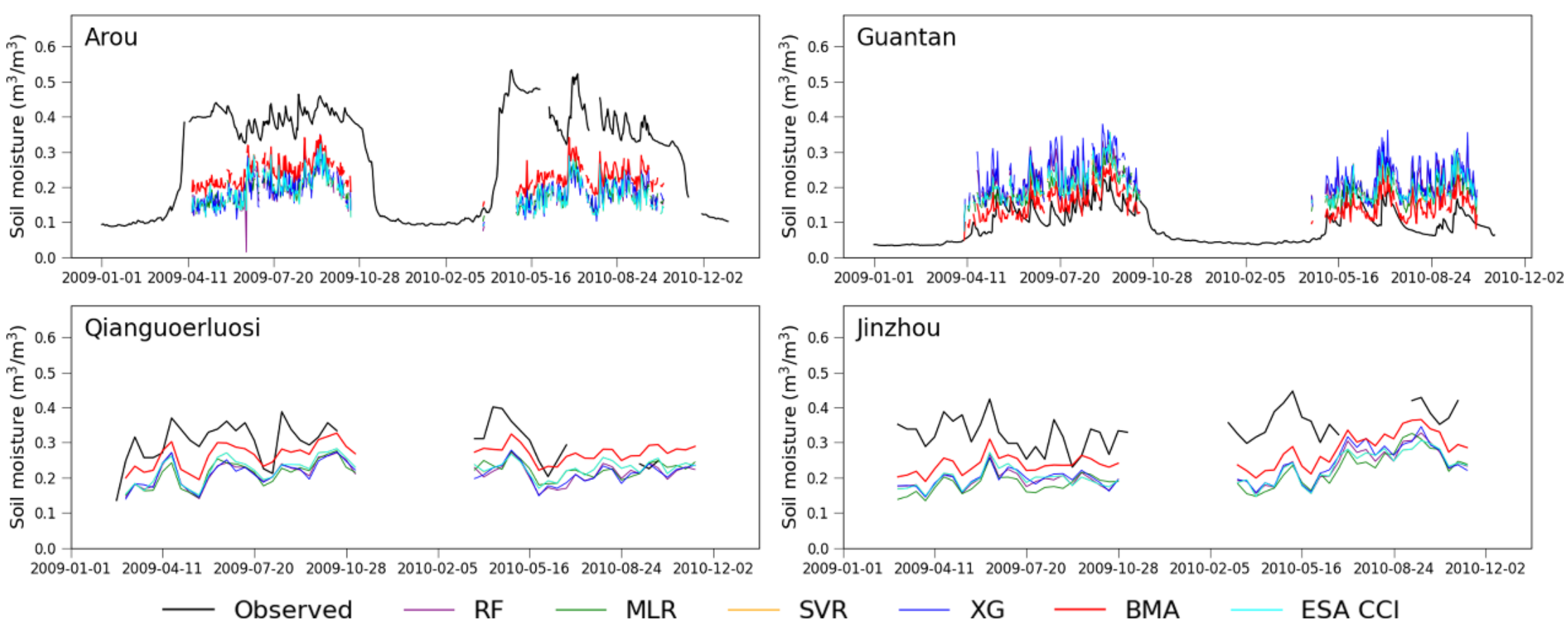

**Figure 9: Time series plots depicting the temporal dynamics of soil moisture, encompassing data from in situ measurements at four selected locations. The time series also include soil moisture values obtained from ESA CCI, as well as downscaled results derived from five machine learning models: random forest (RF), multiple linear regression (MLR), support vector regression**

**(SVR), extreme gradient XG Boost (XG), and Bayesian model averaging (BMA). The temporal resolution of the model data at Arou and Guantan stations is 1 day, while Qianguoerluosi and Jinzhou stations have a resolution of 10 days to match that of the observational data.**

### 4.4 Cross validation with numerical model outputs

The ERA5 and Noah surface soil moisture datasets are chosen to compare with BMA downscaling results, given their efficacy and extensive adoption (Li et al., 2020; Yuling et al., 2022). As illustrated in Fig. 10, the R and MAE distributions of the ERA5 data within the study area and the Noah data within the Loess Plateau are utilized. Results reveal that the BMA ensemble outcomes exhibit reasonable performance in terms of higher R values and lower MAE values when compared to both the ERA5 and Noah datasets. The validation outcomes with the ERA5 data exhibit consistency across most regions, reflecting enhanced R and diminished MAE. In specific regions of the west-central part of the northern China, slight deviations are apparent, characterized by lower R and comparatively higher MAE. Concerning the Noah dataset, the R values remain high, and the MAE values reduced across a substantial portion of the Loess Plateau. A marginal subset of the north-central and south-western parts of Loess Plateau exhibit higher MAE and a slightly diminished R correlation. Collectively, the alignment between the BMA ensemble results and both the ERA5 and Noah datasets underscores the credibility of the downscaled soil moisture derived from our model.

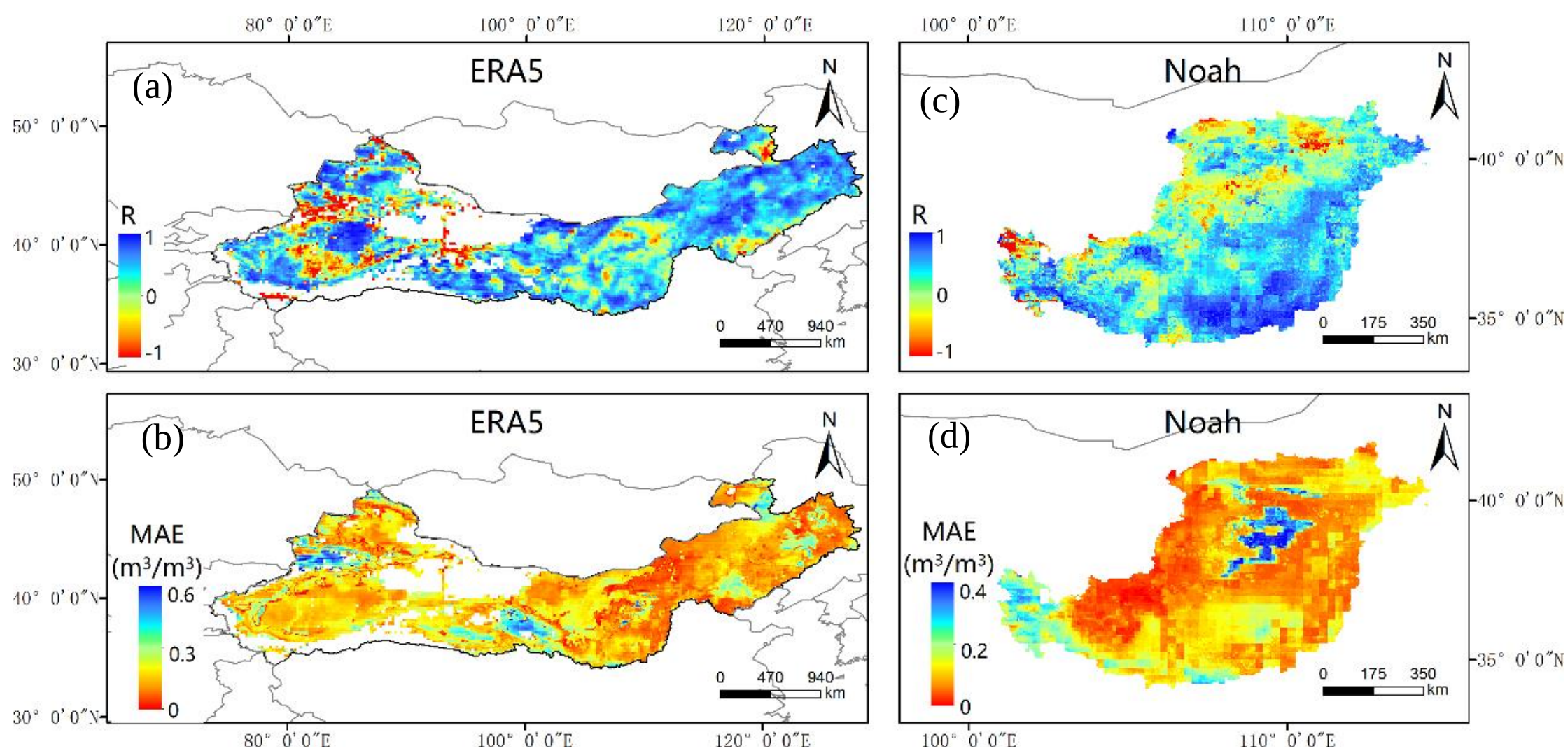

**Figure 10: Assessment of downscaled soil moisture against larger-scale numerical models. Comparative evaluation of the accuracy of 1-km downscaled soil moisture in contrast to 10-km ERA5 products, depicted through (a) correlation coefficient (R) and (b) mean absolute error (MAE). The evaluation extends to 6-km Noah simulations, as shown in (c) for R and (d) for MAE.**

### 4.5 Assessment in terms of drought monitoring

To assess the capability of soil moisture products in capturing drought dynamics, two drought-affected regions (Fig. S5) are identified based on PDSI values equal to or lower than -2 and drought periods exceeding 80% (Long et al., 2014). In each of

these identified regions, we compute the temporal averages of all station measurements, alongside the BMA ensemble results and the ESA CCI data, for the stations positioned within the selected region (Fig. 11). Evidently, the time series plots for both identified regions underscore the concordance between the BMA ensemble outcomes and the ESA CCI data with the in-situ observations. This alignment is conspicuous in their representation of fluctuations in soil moisture levels. Moreover, the characterization of instances of severe drought conditions is consistent with the in-situ measurements. Hence, it can be inferred that the BMA results capture and reflect the prevailing soil drought conditions.

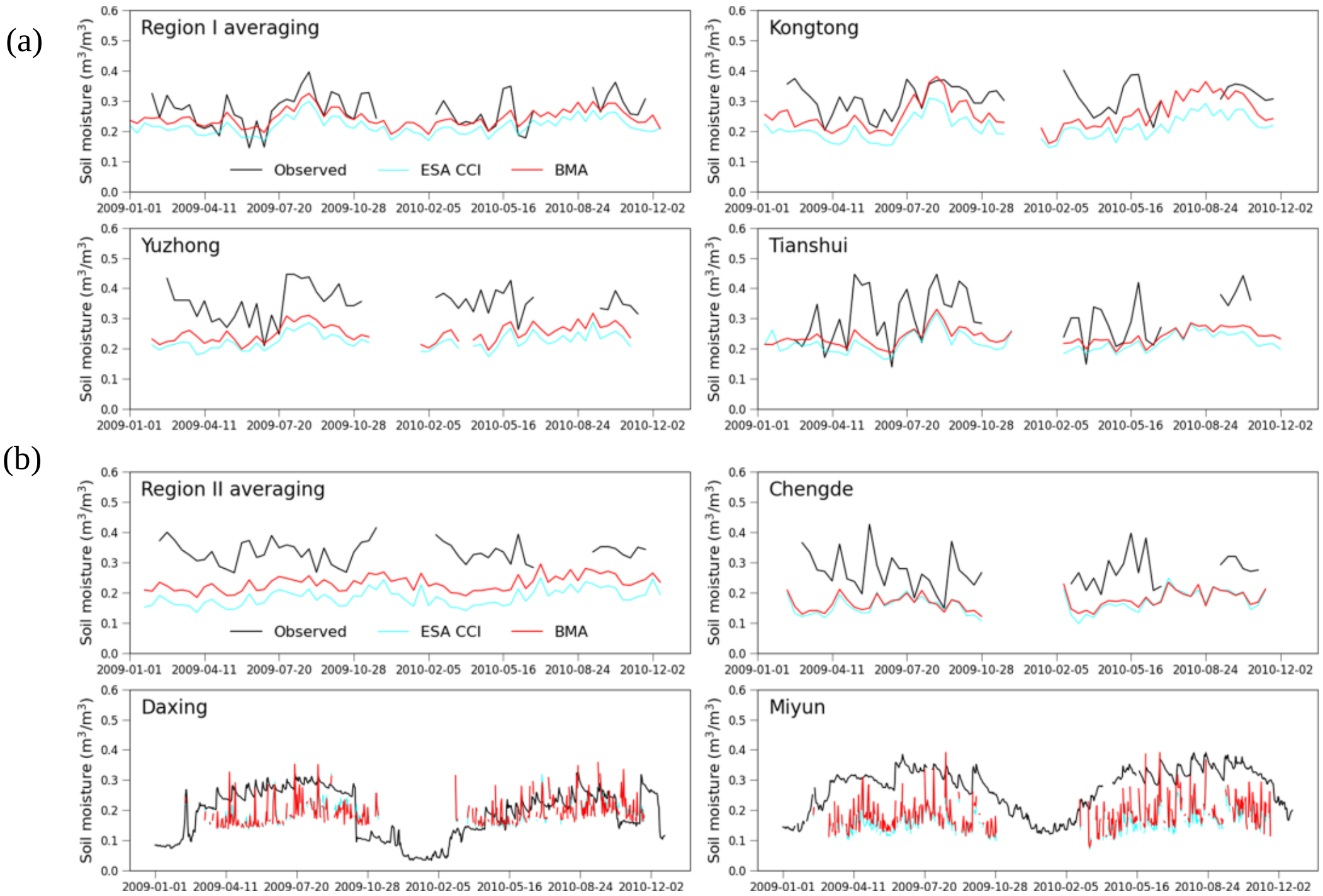

**Figure 11: Temporal analysis of soil moisture. (a) Comparative time series of ESA CCI and BMA downscaled soil moisture, focusing on the regional average within Region I. Additionally, the temporal trends are presented for three specific stations located within this region. (b) Similar for Region II, with attention to the regional average as well as individual stations within this region. The temporal resolution of the model data at Daxing and Miyun stations is 1 day, while other stations have a resolution of 10 days to match that of the observational data.**

### 4.6 Uncertainty analysis

A leave-one-out analysis is conducted to assess the impacts of explanatory variables on model performance. Each input variable is excluded one at a time, and the BMA-integrated results are validated against ground station measurements. Fig. 12(a) shows the validation outcomes following the exclusion of corresponding variables (i.e., albedo, NDVI, precipitation, LST, and DEM), as well as that incorporates all parameters. The removal of an individual parameter has a substantial but controlled effect on the model performance, generally preserving the trends. In comparison to the BMA model including all parameters, the exclusion of NDVI leads to an obvious decrement in R. This observation underscores the relatively higher

impact of NDVI among these parameters on the model outcomes, mostly attributing to NDV's capacity to encapsulate surface vegetation state and coverage. The exclusion of precipitation and DEM results in a perceptible decrease in accuracy for their respective validation outcomes (Park et al., 2017b).

The performance of machine learning models heavily relies on the quantity and quality of the training samples utilized (Géron, 2022). To explore such impact, an analysis is conducted, i.e., a stepwise removal of training samples—comprising 10%, 20%, 30%, 40%, and 50% of the BMA model training dataset. Fig. 12(b) shows the discernible but not strong fluctuations upon the sequential removal of samples. This can be attributed to the substantial volume of samples integrated into the model, rendering the impact of the removal of a portion of samples comparatively minor. This lends support to the stability of the training samples employed in the model, underscoring their representative nature and consistent influence on the model's performance.

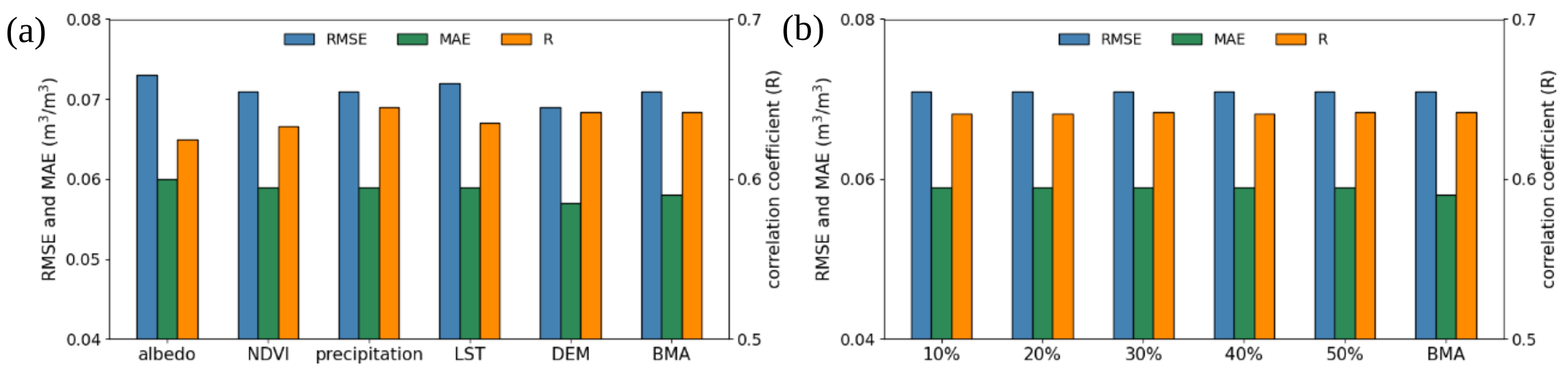

**Figure 12: Model uncertainty. (a) Verification of model robustness through leave-one-out analysis. The subplots illustrate the validation accuracy of BMA downscaled soil moisture upon removal of individual input parameters. The rightmost plot presents results when all parameters are retained. (b) Assessment of training sample impact. The subplots show validation accuracy with varying percentages (10%, 20%, 30%, 40%, 50%) of BMA training samples removed, along with the scenario when all samples are included.**

Additionally, our model excels in capturing intermediate soil moisture levels, yet it tends to underestimate high values and overestimate low values within the soil moisture range. This indicates that the machine learning model effectively represents dominant factors but may overlook subtle signals crucial for capturing extreme values. To emphasize this asymmetric performance in downscaling, we check the model residuals and find no discernible pattern. Such asymmetry is common in machine learning models, partly attributed to limited data availability in extreme regimes (Liu et al., 2023; Sadayappan et al., 2022). This uneven performance is particularly pronounced over dry regions, where lower accuracy is generally observed due to uncertainties in the available training sites and model applicability.

### 4.7 Model merit and shortcoming

Our study is dedicated to enhancing the spatial resolution and hydrological delineation of satellite-based soil moisture retrievals across the Three Northern region of China. These regions, characterized by intricate landscape diversity and climatic dynamics, demand an advanced but robust approach to capture the complexities inherent in their hydrological

response. The uniqueness of our approach lies in its integration of robust methodologies, strategically chosen to maximize the accuracy of soil moisture. Collectively, our framework produces more accurate and coherent high-resolution soil moisture data compared to conventional single-method downscaling approaches, and this superiority can be attributed to three key aspects. i) By employing clustering analysis, we group similar spatial and temporal contexts before model fitting. This enables the model to extract meaningful patterns from the extensive dataset. Our results show that the cluster-based model outperforms models without clustering (Fig. S6). The observed improvement in accuracy, approximately 5%, is expected as clustering into more homogeneous groups allows access to greater variance in soil moisture, thereby enhancing the bias-variance trade-off for complete scenes (Merentitis et al., 2014; Xiao et al., 2018a). Additionally, while our clustering analysis based on soil moisture data offers valuable insights, it may neglect other influential factors like vegetation types and terrain, possibly constraining model accuracy. Future studies should consider partition modeling incorporating complex ecosystem characteristics to improve the spatial representation of soil moisture across diverse regions and ecosystems. ii) The adaptive spatial-temporal window strategy ensures that our model adapts to the specific characteristics of different regions, accommodating variations in soil moisture dynamics across diverse terrains. This strategy addresses issues related to neglecting temporal information in image sequences and the low contrast between objects and background (Liu et al., 2020; Mahadevan and Vasconcelos, 2010). This framework potentially outperforms global models that downscale soil moisture from entire images, especially in handling extreme values. Specifically, the relative model bias across different quantiles (Fig. S7) indicates that our proposed models exhibit 7% less bias compared to the global model, and this improvement could reach approximately 11% in the 90% soil moisture quantile. iii) Another merit of our work is the fusion of soil moisture downscaling outputs derived from four distinct machine learning techniques, harmonized through Bayesian modeling. This strategic integration leverages the diverse strengths of individual methods, resulting in a reliable estimation of soil moisture. The Bayesian model utilizes prior knowledge from in-situ soil moisture across the entire study region to constrain the general dynamics of downscaled soil moisture. The original temporal pattern of the series is largely retained due to this constraint, allowing for the observation of different dynamics between downscaled soil moisture and in situ soil moisture (Ramoni et al., 2002). The weights assigned to individual models exhibit noticeable differences, with random forest receiving higher weights ($0.38\pm0.14$). However, other approaches also contribute substantially, as evidenced by weights larger than 0.13 (Fig. S8). This diversity in weights reflects the effective constraint of downscaled soil moisture by the multiple ensembles to the observations.

To substantiate the efficacy of our proposed methodology, we benchmark our model results against those of earlier studies. The summarized comparative analysis in Table 5 highlights the reasonable performance of our approach, showing its potential to match or even surpass existing techniques in terms of accuracy. Traditional studies generally encounter challenges in achieving fine spatial resolution downscaled soil moisture over large scales, especially in drylands and critical zones. This challenge often arises due to the underutilization of available knowledge related to model structure and cloud contamination. The soil moisture products derived from prior studies may fall short for applications such as regional water resources management, drought event diagnosis, and land-atmosphere feedback analysis. Our framework, which leverages

multiple sources of knowledge and in-situ observations, proves well-suited for addressing the intricate soil moisture downscaling requirements prevalent in complex geographical regions.

**Table 5.** Comparison of our model with other studies

| ID | Methods | Study Area | Data Used | R/$R^2$ | RMSE/ ubRMSE ($m^3/m^3$) | Reference Studies |
|---|---|---|---|---|---|---|
| 1 | GWR | Yangtze and Huaihe rivers in China | AMSR-2, MODIS | R = 0.54-0.55 | ubRMSE = 0.074 | Song et al. (2019) |
| | UTF | | | R = 0.28-0.37 | ubRMSE = 0.097-0.101 | |
| 2 | RF | South Korea/Australia | AMSR-E, MODIS | R = 0.71-0.84 | RMSE = 0.049-0.057 | Im et al. (2016) |
| | Boosted regression trees | | | R = 0.75-0.77 | RMSE = 0.052-0.078 | |
| | Cubist | | | R = 0.70-0.61 | RMSE = 0.051-0.063 | |
| 3 | MATCH | Qinghai-Tibet Plateau | SMAP, MODIS | R = 0.55 | ubRMSE = 0.047 | Shangguan et al. (2023) |
| 4 | CART | Northeastern China | ESA CCI, MODIS | $R^2$ = 0.135 | RMSE = 0.076 | Liu et al. (2018) |
| | KNN | | | $R^2$ = 0.130 | RMSE = 0.074 | |
| | BAYE | | | $R^2$ = 0.081 | RMSE = 0.075 | |
| | RF | | | $R^2$ = 0.191 | RMSE = 0.073 | |
| 5 | Our Bayesian Ensemble | Three North Regions | ESA CCI, MODIS | R = 0.342–0.642 | RMSE = 0.071-0.169 | |

While our model demonstrates commendable performance, it is essential to acknowledge its shortcomings. Firstly, the distribution of validation stations predominantly favors the eastern study area, potentially compromising the generalizability of validation outcomes across the broader region. Future work should be conducted to collect more measurements from western stations to enhance the representativeness of the validation. Secondly, the selection of four regression methods in this study influences downscaling outcomes, and exploring alternative methods that may display superior results remains a viable avenue for future research. Lastly, our choice of employing the Bayesian model for integrating different downscaling outcomes may benefit from considering alternative ensemble methods in forthcoming research. These constraints highlight the need for refining our model's efficacy and applicability in soil moisture downscaling across expansive and varied terrains.

## 5 Conclusion

The emergence of remote sensing technology has enabled extensive soil moisture monitoring, yet prevailing satellite products commonly face limitations in restricted spatial resolution, hindering their broader utility, especially across large-

scale regions. In this study, we establish a robust framework for achieving high-resolution soil moisture by leveraging an ensemble machine learning approach alongside diverse knowledge sources. This framework is implemented to downscale ESA CCI soil moisture, converting it from a resolution of 25 km to 1 km, employing ancillary soil moisture-related data, an adaptive spatial-temporal window strategy, and four distinct machine learning techniques. Crucially, the integration of outcomes from these machine learning methods using a Bayesian model enhances reliability and coherence in the downscaled soil moisture datasets, providing insights to the remote sensing and eco-hydrology field.

Our study spans the expansive Three Northern region of China, covering more than five million square kilometers. Validation against three distinct ground station datasets underscores the efficacy of our model, affirming its potential to enhance the precision of the initial ESA CCI soil moisture. In contrast to single machine learning techniques, our model combines the strengths of each, resulting in more coherent and precise outcomes. Cross-validation with ERA5 and Noah soil moisture data further reinforces the reliability of our model's performance. Furthermore, the downscaled outputs show promise for use in drought assessment, particularly within the arid and semi-arid regions of northern China susceptible to drought and water stress.

In conclusion, our proposed framework facilitates soil moisture downscaling, demonstrating both reliability and scientific soundness. This serves as a reference for acquiring high-accuracy, high-resolution, and expansive-scale soil moisture data, with implications for diverse domains, including water resources management, drought monitoring, and crop yield estimation.

**Code/Data availability**

All the datasets used in this study are open to the public. The ESA CCI soil moisture dataset and ERA-5 reanalysis datasets is collected from the European Centre for Medium-range Weather Forecasts (ECMWF). The National Aeronautics and Space Administration team provides the MODIS products, GPM data and SRTM DEM data. The China Watershed Allied Telemetry Experimental Research (WATER) project, Chinese Ecosystem Research Network (CERN), and Maqu soil moisture monitoring network provides available in situ measurements at the website http://data.tpdc.ac.cn/en/. The Chinese regional ground meteorological dataset is collected from the National Tibetan Plateau Data Center (http://data.tpdc.ac.cn).

**Author contribution**

Kai Liu, Hongyan Zhang and Xueke Li designed the theoretical formalism. Hongyan Zhang and Yong Bo developed the model code and performed the simulations. Dehui Li and Long Li performed the analytic calculations. Shudong Wang and Xueke Li supervised the study. Shudong Wang, Xueke Li and Hang Li contributed to the final version of the paper.

## Competing interests

The contact author has declared that neither they nor their co-authors have any competing interests.

## Acknowledgements

This study was supported by the National Natural Science Foundation of China (42141007), the Key Special Project for the Action of "Revitalizing Mongolia through Science and Technology" (2022EEDSKJXM003).

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
