# Peer review of "Downscaling satellite-derived soil moisture in the Three North region using ensemble machine learning and multiple-source knowledge integration"

_Hydrology and Earth System Sciences, 2024_

## Author Comment (AC1)

**Response to RC1:**

The major contribution of this paper is the use of Bayesian Model Averaging (BMA) to combine the outputs from several different empirical ("machine learning") techniques for soil moisture downscaling. The authors test this methodological innovation by comparing to a large dataset of in-situ soil moisture sensors scattered across northern China. I have several comments that I hope the authors will address.

Response: Thank you for your insightful and constructive feedback, which has been invaluable in improving the quality of our manuscript. We have carefully addressed and responded to each comment in detail. Furthermore, we have introduced additional independent methods and regional comparisons to strengthen the robustness of our analysis and broaden the scope of our findings.

1. First, I believe that statistical derivation of BMA assumes that the models are independent of each other. It seems like the models developed here are likely not independent because they have been developed using the same inputs and the same training data. Have the authors tested whether this assumption applies to their models? If they are dependent, what is the impact on the results?

Response: We appreciate this comment and recognize that BMA indeed assumes independence among models. However, achieving complete independence among models can be challenging in practice, especially when models rely on the same meteorological input data. In our study, we found Pearson correlations among the four models' outputs to be approximately 0.75-0.85, indicating moderate dependency.

While this dependency might influence the BMA assumption of model independence, it does not necessarily violate it. Generally, BMA's independence assumption is not an absolute requirement for zero correlation but rather seeks models that contribute some level of unique information, thus reducing uncertainty in the ensemble [1, 2]. In practice, even with shared inputs, if the models exhibit unique structural differences such as ET and soil moisture [3, 4], Bayesian model can still effectively enhance ensemble accuracy by drawing on these differences.

Additionally, the observed dependencies in our models resemble convergence toward the true target rather than redundancy, meaning that each model's output aims to approximate the same objective (i.e., true soil moisture values) rather than merely replicating each other. As long as models are not entirely redundant (providing the same information), BMA can still yield effective integration and improvements in overall accuracy.

To further explore the impact of dependency, we also conducted additional analyses, which have been added in the main text section 4.5 and supplementary Fig. S8.

1) We assessed the sensitivity of the BMA ensemble by systematically removing one model at a time to evaluate any significant changes in accuracy. We observed accuracy reductions ranging from 7-12% when specific models were removed. This outcome suggests that each model contributes unique information that significantly impacts the ensemble's performance. If the models were highly dependent, removing one would not cause a notable change in accuracy, as the remaining models would effectively compensate. This sensitivity analysis indicates that while the datasets are correlated, they are not redundant, as each provides distinct characteristics or

features. The purpose of BMA is to leverage unique information from each model through weighted averaging to reduce uncertainty. If there were strong dependencies, BMA's weighting mechanism would be less effective. However, the observed impact on accuracy confirms that these dependencies do not compromise BMA's integrative performance. Therefore, this analysis reasonably demonstrates that any interdependence among the models has only a limited impact on BMA's effectiveness, and that each model contributes valuable information to the ensemble.

2) To further investigate the impact of dependencies, we also compared BMA results with those from a Hierarchical Bayesian Model (HBM) [5, 6], which explicitly accounts for dependency structures. HBM incorporates each data source as a distinct hierarchical level, introducing a "data source bias" random effect to model deviations of each source from the global mean. This framework allows HBM to account for dependencies by quantifying and mitigating inter-model biases. Our results showed that HBM achieved an accuracy improvement of less than 8% over BMA, indicating that while HBM accounts for dependencies, these dependencies and any associated systematic biases only moderately impact ensemble accuracy. This suggests that BMA remains a viable approach for practical applications, particularly in scenarios where data complexity or computational constraints make it preferable. Although Bayesian methods like HBM offer advantages over traditional techniques, they are sensitive to prior settings. The choice of priors can strongly influence posterior distributions, and if observational data does not fully represent true conditions—due to factors such as low data quality or limited spatial coverage— model calibration may affect overall accuracy.

[Figure]

*Figure S8. Sensitivity analysis of the Bayesian Model Averaging (BMA) ensemble, evaluating the impact of systematically removing one model at a time on accuracy. Accuracy reductions of 7–12% were observed when specific models were excluded, indicating that each model contributes unique information critical to the ensemble's performance. If the models were highly dependent, removing one would result in minimal accuracy changes, as the remaining models would compensate. This analysis demonstrates that while the datasets exhibit some correlation, they are not redundant, as each provides distinct and valuable features. To further explore the impact of model dependencies, we compared the BMA results with those from a Hierarchical Bayesian Model (HBM) (Sairam et al., 2019), which explicitly incorporates dependency structures. HBM treats each data source as a distinct hierarchical level and introduces a "data source bias" random effect to account for deviations of individual sources from the global mean. This approach enables HBM to quantify*

*and mitigate inter-model biases. Results showed that HBM achieved a modest accuracy improvement of less than 8% over BMA, suggesting that while HBM better accounts for dependencies, these dependencies and associated systematic biases have only a moderate effect on ensemble accuracy. These findings underscore BMA's robustness and practicality, particularly in scenarios where data complexity or computational constraints make it preferable. While advanced Bayesian approaches like HBM offer benefits, such as explicitly modeling dependencies, they are sensitive to prior settings. The choice of priors can significantly influence posterior distributions, and inaccuracies in observational data—stemming from low quality or limited spatial coverage—can impact model calibration and overall accuracy.*

[1] Hollenbach, F. M., & Montgomery, J. M. (2020). Bayesian model selection, model comparison, and model averaging. *The sage handbook of research methods in political science and international relations*, 937-960.
[2] Raftery, A. E., Gneiting, T., Balabdaoui, F., & Polakowski, M. (2005). Using Bayesian model averaging to calibrate forecast ensembles. *Monthly weather review*, *133*(5), 1155-1174.
[3] Shao, X., Zhang, Y., Liu, C., Chiew, F. H., Tian, J., Ma, N., & Zhang, X. (2022). Can indirect evaluation methods and their fusion products reduce uncertainty in actual evapotranspiration estimates?. *Water Resources Research*, *58*(6), e2021WR031069.
[4] Chen, Y., Yuan, H., Yang, Y., & Sun, R. (2020). Sub-daily soil moisture estimate using dynamic Bayesian model averaging. Journal of Hydrology, 590, 125445.
[5] Kim, T. J., Kwon, H. H., & Lima, C. (2018). A Bayesian partial pooling approach to mean field bias correction of weather radar rainfall estimates: Application to Osungsan weather radar in South Korea. *Journal of Hydrology*, *565*, 14-26.
[6] Sairam, N., Schröter, K., Rözer, V., Merz, B., & Kreibich, H. (2019). Hierarchical Bayesian approach for modeling spatiotemporal variability in flood damage processes. Water resources research, 55(10), 8223-8237.

2. Second, all the downscaling methods considered provide very little improvement in the soil moisture estimates. A key goal of downscaling is to include fine scale spatial variability that is not present in the coarse resolution input. However, when I examine the histograms in Figure 6, I see no increase in the variability of soil moisture when the downscaling methods are applied. Some of the methods have less variability than the coarse resolution input. Are these methods successfully introducing any variability in the patterns? Also, the accuracy of the BMA method is only slightly better than the coarse resolution input. The exact improvement is difficult to see because Table 4 does not include the performance of the coarse resolution input nor the overall performance across all the datasets used. Those should be added). The authors seem satisfied with the improvement in their discussion and conclusions, but it seems like the improvements do not warrant the huge processing involved. The authors consider relatively few variables. Could better performance be achieved by using model inputs?

Response: Thank you for your constructive feedback, which has strengthened our study. 1) Our study's primary objective was to produce a 1-km soil moisture product by downscaling satellite-derived datasets and calibrating model bias with ground-based soil moisture measurements. This approach differs from traditional downscaling. Generally, in the remote sensing field, downscaling refers to using coarse-resolution data along with high-resolution auxiliary datasets to predict the target variable at finer resolutions. The primary focus here was not solely on increasing spatial variability but on generating reliable high-resolution soil moisture data for arid regions such as Northern China, where coarse-resolution ESA CCI data may overestimate surface moisture due to the limitations in capturing finer-scale processes [1, 2]. In arid and semi-arid regions, high surface exposure and low vegetation cover lead to rapid surface drying. Consequently, ESA CCI's microwave sensing, while sensitive to surface moisture, struggles to accurately capture deeper moisture levels, resulting in potential overestimation of surface soil moisture.

Additionally, machine learning techniques, particularly those constrained by field observations, tend to smooth extreme values, resulting in reduced spatial variability [3, 4]. Current machine learning models often exhibit limitations in capturing extreme soil moisture values, especially under drought conditions, and tend to make conservative predictions, which can lead to underestimation in dry areas.

It's important to clarify that a narrower range in soil moisture values after downscaling does not imply a reduction in spatial variability [5-7]. The spatial variability depends on the model's capability to capture local details and the heterogeneity present in the original data. Downscaling to higher resolution allows local features—such as topography, land cover, or moisture gradients—to emerge more clearly, thereby enhancing spatial variability where appropriate. In fact, as observed in the histograms and box plots, our downscaled product shows increased clustering in the middle range, with values in this range trending upward after downscaling. Using metrics like the coefficient of variation (CV) and Moran's I, we observe an increase in local spatial variability within the middle range, while the extreme ranges exhibit less pronounced variability.

In the new version, we have added the related context in the main text section 4.1 and supplementary Table S3.

*Table S3. Coefficient of variation (CV) and Moran's I index*

| | ESA CCI | RF | MLR | SVR | XG | BMA |
|---|---|---|---|---|---|---|
| CV | 0.321 | 0.328 | 0.313 | 0.324 | 0.332 | 0.324 |
| Moran's I [0-100%] | 0.994 | 0.964 | 0.955 | 0.994 | 0.946 | 0.972 |
| Moran's I [0-15%] | 0.991 | 0.755 | 0.876 | 0.991 | 0.846 | 0.86 |
| Moran's I [15-85%] | 0.99 | 0.795 | 0.922 | 0.989 | 0.825 | 0.803 |
| Moran's I [85-100%] | 0.991 | 0.864 | 0.921 | 0.991 | 0.895 | 0.902 |

*Note: CV measures overall variability, with higher values indicating stronger heterogeneity. Moran's I quantifies spatial distribution patterns, where higher values reflect weaker heterogeneity and stronger spatial autocorrelation. The CV is calculated for the entire dataset. Moran's I index is determined using a simple four-neighborhood relationship, with brackets indicating different sample divisions. The 0-100% range represents the full sample, while the 0-15%, 15-85%, and 85-100% ranges correspond to low-value, mid-range, and high-value distributions, respectively.*

*Table 4. Comparison of BMA and individual machine learning*

| Stations | Num | R | | | | | | | | | | | |
|---|---|---|---|---|---|---|---|---|---|---|---|---|---|
| | | RF | | MLR | | SVR | | XG | | BMA | | ESA CCI | |
| | | All | Mid | All | Mid | All | Mid | All | Mid | All | Mid | All | Mid |
| NZW | 5044 | 0.323 | 0.383 | 0.338 | 0.411 | 0.321 | 0.398 | 0.325 | 0.399 | 0.342 | 0.424 | 0.321 | 0.375 |
| CERN | 263 | 0.567 | 0.664 | 0.617 | 0.693 | 0.629 | 0.705 | 0.573 | 0.672 | 0.642 | 0.721 | 0.586 | 0.647 |
| QXZ | 1204 | 0.477 | 0.571 | 0.480 | 0.593 | 0.478 | 0.583 | 0.468 | 0.554 | 0.514 | 0.610 | 0.479 | 0.531 |

| Stations | Num | RMSE (m3/m3) | | | | | | | | | | | |
|---|---|---|---|---|---|---|---|---|---|---|---|---|---|
| | | RF | | MLR | | SVR | | XG | | BMA | | ESA CCI | |
| | | All | Mid | All | Mid | All | Mid | All | Mid | All | Mid | All | Mid |
| NZW | 5044 | 0.138 | 0.108 | 0.136 | 0.105 | 0.138 | 0.106 | 0.137 | 0.104 | 0.137 | 0.097 | 0.138 | 0.115 |
| CERN | 263 | 0.073 | 0.060 | 0.073 | 0.059 | 0.073 | 0.060 | 0.075 | 0.061 | 0.071 | 0.054 | 0.084 | 0.064 |
| QXZ | 1204 | 0.169 | 0.125 | 0.169 | 0.131 | 0.168 | 0.132 | 0.169 | 0.128 | 0.169 | 0.115 | 0.168 | 0.139 |

| Stations | Num | MAE (m3/m3) | | | | | | | | | | | |
|---|---|---|---|---|---|---|---|---|---|---|---|---|---|
| | | RF | | MLR | | SVR | | XG | | BMA | | ESA CCI | |
| | | All | Mid | All | Mid | All | Mid | All | Mid | All | Mid | All | Mid |
| NZW | 5044 | 0.114 | 0.092 | 0.113 | 0.091 | 0.114 | 0.095 | 0.114 | 0.095 | 0.114 | 0.089 | 0.114 | 0.101 |
| CERN | 263 | 0.060 | 0.038 | 0.061 | 0.039 | 0.061 | 0.039 | 0.062 | 0.040 | 0.058 | 0.032 | 0.067 | 0.045 |
| QXZ | 1204 | 0.157 | 0.119 | 0.158 | 0.123 | 0.156 | 0.117 | 0.158 | 0.120 | 0.158 | 0.114 | 0.157 | 0.134 |

Note: "All" refers to the full set of sample points, whereas "Mid" denotes the subset of sample points that fall within the 15-85% range.

2) We have revised Table 4 to include performance metrics for the coarse-resolution input, enabling clearer comparison across all datasets. While accuracy validation is crucial, it represents just one aspect of our evaluation. Site-scale validations are subject to scale effects, so we conducted a comprehensive assessment that included drought event capture and product comparisons. Our Bayesian framework, combined with ground observations, successfully generated a stable high-resolution soil moisture dataset.

Although the overall accuracy gains may appear modest due to the large study area and site data scale effects, our work remains unique. Few studies attempt site-based soil moisture downscaling over a large area such as northern China. When focusing on specific regions, such as the Loess Plateau and the North China Plain—semi-arid areas with rich site data—the accuracy improvements become more pronounced, highlighting the robustness and utility of our dataset and approach.

3) Our choice of explanatory variables was guided by two main principles: (i) ensuring that we had stable, reliable remote sensing observations available at a large scale, thus allowing for future applications in other regions or even at a global scale; and (ii) selecting variables with strong correlations to soil moisture but minimal redundancy. The five variables we chose represent key drivers of soil moisture across meteorological, ecological, and hydrological dimensions, with low inter-correlation. Related context has been improved in the main text section 3.1 and Figure 3.

[Figure]

*Figure 3: Assessment of explanatory variables' feasibility. (a) Average (blue bar) and standard deviation (error bar) of permutation-based importance of explanatory variables concerning soil moisture. (b) Average Pearson correlation coefficients among different explanatory variables, including correlations with two independent soil moisture data sources.*

Other potential variables, such as vegetation indices (e.g., EVI), downwelling radiation, and evapotranspiration, were excluded due to their high correlation with the selected variables, limited efficacy in arid and semi-arid regions like northern China, and inconsistency in accuracy at daily and fine spatial scales. Soil attributes, such as texture and classification, are often critical in soil moisture modeling [8], yet in our study, they contributed less than 2% to overall accuracy improvements. This may be due to the relative homogeneity in soil texture across northern China, where sandy soils and loams predominate, offering little spatial variation to capture soil moisture heterogeneity. Moreover, the spatial partitioning employed before model implementation likely accounted for soil characteristics within each subregion, further diminishing the impact of texture. Consequently, soil texture added minimal explanatory value. In summary, while our choice of variables may omit certain minor features, the overall accuracy is robust and serves as a valuable reference for large-scale and global soil moisture studies.

[1] Zhang, G., Su, X., Ayantobo, O. O., & Feng, K. (2021). Drought monitoring and evaluation using ESA CCI and GLDAS-Noah soil moisture datasets across China. *Theoretical and Applied Climatology*, *144*, 1407-1418.
[2] Dorigo, W. A., Gruber, A., De Jeu, R. A. M., Wagner, W., Stacke, T., Loew, A., ... & Kidd, R. (2015). Evaluation of the ESA CCI soil moisture product using ground-based observations. *Remote Sensing of Environment*, *162*, 380-395.
[3] Sadayappan, K., Kerins, D., Shen, C., & Li, L. (2022). Nitrate concentrations predominantly driven by human, climate, and soil properties in US rivers. *Water Research*, *226*, 119295.
[4] Bo, Y., Li, X., Liu, K., Wang, S., Li, D., Xu, Y., & Wang, M. (2024). Hybrid theory-guided data driven framework for calculating irrigation water use of three staple cereal crops in China. *Water Resources Research*, *60*(3), e2023WR035234.
[5] Wang, F., & Tian, D. (2024). Multivariate bias correction and downscaling of climate models with trend-preserving deep learning. *Climate Dynamics*, *62*(10), 9651-9672.
[6] Maraun, D., Wetterhall, F., Ireson, A. M., Chandler, R. E., Kendon, E. J., Widmann, M., ... & Thiele-Eich, I. (2010). Precipitation downscaling under climate change: Recent developments to bridge the gap between dynamical models and the end user. *Reviews of geophysics*, *48*(3).
[7] Latombe, G., Burke, A., Vrac, M., Levavasseur, G., Dumas, C., Kageyama, M., & Ramstein, G. (2018). Comparison of spatial downscaling methods of general circulation model results to study climate variability during the Last Glacial Maximum. *Geoscientific Model Development*, *11*(7), 2563-2579.
[8] Xu, M., Yao, N., Yang, H., Xu, J., Hu, A., de Goncalves, L. G. G., & Liu, G. (2022). Downscaling SMAP soil moisture using a wide & deep learning method over the Continental United States. *Journal of Hydrology*, *609*, 127784.

3. Third, little consideration is given as to whether the in situ dataset adequately captures 1-km spatial variations in soil moisture (which is the stated goal of the downscaling method). The measurement support is likely very small and the spacing is likely much larger than 1-km. Even

if the downscaling models reproduce this dataset exactly, have we really developed an accurate 1-km resolution soil moisture estimate? Can the authors provide some support that that a given in situ soil moisture observation is representative of its 1-km grid cell? Also, can the authors show that the collection of 1-km grid cells that have in situ observations capture the range of conditions that occur within the region? I believe some support along these lines would greatly strengthen the paper.

Response: Thanks for pointing out this issue.

1) We agree that scale effects are among the most significant challenges in remote sensing validation, especially for soil moisture downscaling to a 1-km resolution. Currently, there is no definitive solution to fully bridge the scale gap between in situ observations and satellite-based products. Capturing soil moisture variability at the 1-km scale is particularly challenging across northern China's extensive 3-million-square-kilometer study area, where diverse climate and surface characteristics further complicate validation. In light of these challenges, our study employs a multi-faceted validation approach. In addition to site-based validation, we incorporate drought event analysis and cross-product comparisons. This broader evaluation framework aligns with mainstream practices in current remote sensing research to address validation limitations from scale effects.

Furthermore, our results from a reduced-sample analysis suggest that the scale effect's impact on model outcomes is less significant than anticipated, supporting the robustness and reliability of our findings despite scale challenges. These approaches together reinforce the credibility of our model outputs by considering spatial variability within the constraints of available data.

2) One of the key strengths of our study is the integration of extensive ground data to calibrate remote sensing products and model outputs, reducing errors arising from surface heterogeneity and better aligning the model with actual ground conditions. However, while this integration helps minimize discrepancies, it can also introduce new mismatches between in situ and satellite data—an area that requires further attention in future research and remains a focus in many recent studies.

In this study, we address regional heterogeneity by dividing the study area into several subregions and calibrating the model with in situ data for each specific subregion. This process allows the model to learn distinct calibration parameters relevant to each area. Although this method effectively incorporates regional variations, it cannot fully eliminate scale-induced transfer effects at finer, localized scales. Moving forward, we plan to explore transfer learning techniques and develop specific loss functions designed to reduce scale-bias when calibrating 1-km satellite data with ground-based measurements. Such methods could enhance calibration accuracy and improve the model's adaptability across different spatial scales.

In the new version, we have added the related context in the main text section 4.5.

3) In response to these concerns, we conducted an additional experiment to examine the impact of scale effects on model accuracy. This experiment focused on the Maqu region [1, 2], located at the transition zone between the Tibetan Plateau and the Loess Plateau. Maqu's relatively flat terrain and predominantly grassland cover makes it suitable for comparative analysis, and the presence of

20 ground stations within a 5x5 grid enhances its suitability as a case study for evaluating scale effects.

Our model showed a significant accuracy improvement in this flat, homogeneous region of Naqu, highlighting the pronounced influence of scale effects in regions with minimal topographic variation. Furthermore, we conducted a sequential data reduction analysis, removing 10%, 20%, 30%, and 40% of ground training data while maintaining the same validation dataset. Although model accuracy was somewhat affected by the reduction in training data, the impact was relatively modest. This finding indicates that while sample data quantity influences the overall outcome, the scale effect on model validation remains relatively minor. Specifically, even with reduced training samples, the validation accuracy remained stable, suggesting that the scale information learned by the model from ground station data is sufficiently generalized to apply to the validation set. In essence, this stability implies that the scale difference between ground station data and 1-km remote sensing data does not introduce significant bias in model validation [3,4].

In the new version, we have added the related context in the main text section 4.5 and supplementary Fig. S7.

[Figure]

*Figure S7. Additional experiment examining the impact of scale effects on model accuracy and the representativeness of in situ datasets in capturing soil moisture spatial variations. (a) The experiment was conducted in the Maqu region, a transitional zone between the Tibetan Plateau and the Loess Plateau, characterized by relatively flat terrain and predominantly grassland cover. (b) These features, combined with the presence of 20 ground stations arranged in a 5x5 grid, make*

*Maqu an ideal case study for evaluating scale effects. (c) The model demonstrated significant accuracy improvements in this flat, homogeneous region, underscoring the pronounced influence of scale effects in areas with minimal topographic variation. A sequential data reduction analysis was also performed, removing 10%, 20%, 30%, and 40% of the ground training data while maintaining the same validation dataset. Although the reduction in training data modestly impacted model accuracy, the effect was relatively minor. The validation accuracy remained stable even with fewer training samples, suggesting that the model effectively generalized the scale information learned from ground station data to the validation set. This stability indicates that the scale differences between ground station data and 1-km remote sensing data introduce negligible bias in model validation, reaffirming the robustness of the model's performance in addressing scale effects.*

[1] Dente, L., Vekerdy, Z., Wen, J., & Su, Z. (2012). Maqu network for validation of satellite-derived soil moisture products. International Journal of Applied Earth Observation and Geoinformation, 17, 55-65.

[2] Liu, K., Li, X., Wang, S., & Zhang, H. (2023). A robust gap-filling approach for European Space Agency Climate Change Initiative (ESA CCI) soil moisture integrating satellite observations, model-driven knowledge, and spatiotemporal machine learning. Hydrology and Earth System Sciences, 27(2), 577-598.

[3] Dorigo, W. A., Gruber, A., De Jeu, R. A. M., Wagner, W., Stacke, T., Loew, A., ... & Kidd, R. (2015). Evaluation of the ESA CCI soil moisture product using ground-based observations. Remote Sensing of Environment, 162, 380-395.

[4] Brocca, L., Hasenauer, S., Lacava, T., Melone, F., Moramarco, T., Wagner, W., ... & Bittelli, M. (2011). Soil moisture estimation through ASCAT and AMSR-E sensors: An intercomparison and validation study across Europe. Remote Sensing of Environment, 115(12), 3390-3408.

4. I would suggest removing the Noah results because they really don't contribute to testing the innovation that is presented.

Response: We acknowledge that capturing soil moisture variability at a 1-km resolution is particularly challenging across northern China's extensive 3-million-square-kilometer study area, where diverse climate and surface conditions further complicate the validation process. Given these challenges, our study employs a multi-faceted validation approach. In addition to site-based validation, we incorporate drought event analysis and cross-product comparisons. This comprehensive evaluation framework aligns with mainstream practices in remote sensing research and is designed to address the limitations posed by scale effects in validating downscaled products.

In response to the feedback from the editor-in-chief and reviewers, we have retained the Noah results but revised this section to clarify its relevance. We also streamlined some parts of the manuscript by moving certain elements of the uncertainty analysis from the appendix to the main text, ensuring a clearer focus on the innovative aspects of our methodology.

---

## Author Comment (AC2)

**Response to RC2:**

Summary: This study uses Bayesian model averaging to model soil moisture at a 1km resolution in the Three North region of China by combining the results from 4 individual machine learning methods. Their model uses 5 datasets of varying resolution (LST: 1km/daily, NDVI: 1km/16d, surface albedo: 0.05°/daily, elevation: 90m and precipitation: 0.1°/daily) as explanatory variables for soil moisture. Their model is trained on the 0.25° ESA CCI COMBINED soil moisture product by resampling the high-resolution predictor variables to the same scale and then applied to a lower 1km resolution using the original predictor datasets. The main finding of the paper is that the pearson R correlation coefficient and the RMSE against in-situ measurements from 3 different networks improve slightly in their new high resolution dataset.

General Comments
1. The use of Bayesian model averaging shows an innovative use of machine learning to train models
Response: Thank you for acknowledging our use of BMA as an innovative approach within the machine learning framework. We greatly appreciate your detailed and constructive feedback, particularly your insights from a statistical perspective, which complement the remote sensing hydrology viewpoint. Your suggestions have significantly enhanced the quality of our manuscript and provided valuable learning opportunities, both for improving this work and for my academic growth as a remote sensing hydrology scholar.

2. Misleading title. The authors use the term 'downscaling' to describe their method and the purpose of this study. In the remote sensing community, downscaling usually refers to using a coarse-grained dataset of some environmental variable (like soil moisture) along with auxiliary datasets available at the target resolution as predictor variables in a model to predict the same environmental variable at a higher (target) resolution. The target variable of the training and the validation is typically also used at the target resolution. What the authors describe in the paper would be better described as model calibration.
Response: Thank you for highlighting this important distinction. We recognize that our use of the term "downscaling" might be confusing within the traditional remote sensing context, where downscaling typically involves interpolating a coarse-resolution dataset to a finer spatial resolution, often through auxiliary data as predictor variables. In contrast, our work is more accurately described as a form of "model enhancement". Our objective is to use ground-based observations to enhance the accuracy of a coarse-resolution soil moisture product, thereby improving its alignment with observed values rather than simply increasing its spatial resolution.

Following your suggestion, we have revised the title to reflect the focus on calibration and accuracy enhancement rather than traditional downscaling. The new title is:
"Enhancing satellite-derived soil moisture in the Three North region using ensemble machine learning and multi-source knowledge integration".

3. There is no clear training-validation split in the modelling. This should always be employed when using machine learning models.

Response: Thank you for highlighting this critical aspect of model evaluation. In the revised manuscript, we have clarified our approach to training-validation splitting, which can be found in main text Section 3.2 and 3.5.

Given the limited availability of ground station data for this study, we employed a 5-fold cross-validation approach to assess model accuracy from 2003 to 2010. By training and validating the model on different data splits, we minimized the potential bias associated with any single partition of data. The cross-validation results were then averaged to provide an overall measure of the model's performance across various validation sets, ensuring robust model generalizability. Specifically, to maintain the temporal characteristics of soil moisture, we implemented a spatially-based random split according to geographical location rather than a time-based split. This approach allows the model to better capture spatial variability in soil moisture while preserving temporal integrity.

For optimizing model performance and preventing overfitting, key hyperparameters were fine-tuned to minimize Root Mean Square Error (RMSE) through a 10-fold cross-validation process. This 10-fold cross-validation, conducted across the period of 2011–2013, ensures that the model's parameters are well-calibrated for accuracy and stability. Specifically, nine-tenths of the spatial sample were randomly allocated for model fitting, with the remaining one-tenth reserved for validation.

4. The authors report a narrowing of the distribution of soil moisture values in the 'downscaled' dataset compared to the original CCI SM, which indicates a loss of information. A higher-resolution dataset should have a wider distribution of values than a low-resolution one, if we assume that the low-resolution data is a (weighted) average of the high-resolution data contained in its boundaries. This follows from the central limit theorem.

The time-series values from the BMA are either consistently higher or lower than all of the underlying model predictions (Figure 9). A weighted average must lie somewhere between its constituent values! This suggests that the authors are making an error in their calculations.

Response: Thank you for highlighting these important points. In the new version, we have added the related context in the main text section 4.1, 4.2 and supplementary Figure S4 and Table S3.

1) Traditional machine learning-based downscaling could indeed result in a narrower distribution range due to its tendency to smooth extreme values and prioritize certain variables, thereby producing more conservative predictions. This effect arises because machine learning models are heavily influenced by the training data distribution. In this study, high soil moisture values represented ~20% of the dataset. With such limited examples of extreme moisture, the model may not effectively capture this range, leading it to rely predominantly on mid-to-low value ranges (Figure S4). This reliance can cause underestimation of higher values, especially in arid and semi-arid regions where the scarcity of high-moisture samples limits the model's ability to generalize to higher soil moisture conditions. Furthermore, many machine learning models tend toward

conservative predictions, balancing overall prediction error by minimizing extreme deviations. This approach, while effective for reducing mean square error, can lead to an underestimation of high values, as the model "pulls" extreme values toward the mean [1, 2]. During downscaling, integrating fine-scale spatial features like topography or vegetation cover can also add complexity, especially in high-moisture regions (e.g., near irrigation zones or rivers). This integration challenge may lead the model to underestimate moisture variability, particularly at the higher end of the range. Despite this, our model does capture mid-range variability effectively, demonstrating that it can still reliably represent spatial heterogeneity.

2) The observed narrowing of the soil moisture range should not necessarily be interpreted as information loss [3-5]. In remote sensing downscaling, a narrower range does not contradict enhanced spatial variability. Downscaling increases spatial resolution, allowing for the capture of localized details and finer distinctions in topography, land cover, and moisture gradients. The resulting increase in spatial variability, despite a narrower value range, is a reasonable outcome driven by the model's capacity to accurately represent regional heterogeneity. Evidence from our histograms and box plots indicates that the downscaled dataset shows greater clustering in local median values, with values displaying an upward trend post-downscaling. Metrics such as the coefficient of variation (CV) and Moran's I index further validate an increase in spatial variability, especially in the mid-range (Table S3). This approach, while conservative on extremes, effectively captures the nuanced spatial variability required for soil moisture analysis at finer resolutions.

*Table S3 Coefficient of variation (CV) and Moran's I index*

|  | ESA CCI | RF | MLR | SVR | XG | BMA |
|---|---|---|---|---|---|---|
| CV | 0.321 | 0.328 | 0.313 | 0.324 | 0.332 | 0.324 |
| Moran's I [0-100%] | 0.994 | 0.964 | 0.955 | 0.994 | 0.946 | 0.972 |
| Moran's I [0-15%] | 0.991 | 0.755 | 0.876 | 0.991 | 0.846 | 0.86 |
| Moran's I [15-85%] | 0.99 | 0.795 | 0.922 | 0.989 | 0.825 | 0.803 |
| Moran's I [85-100%] | 0.991 | 0.864 | 0.921 | 0.991 | 0.895 | 0.902 |

*Note: CV measures overall variability, with higher values indicating stronger heterogeneity. Moran's I quantifies spatial distribution patterns, where higher values reflect weaker heterogeneity and stronger spatial autocorrelation. The CV is calculated for the entire dataset. Moran's I index is determined using a simple four-neighborhood relationship, with brackets indicating different sample divisions. The 0-100% range represents the full sample, while the 0-15%, 15-85%, and 85-100% ranges correspond to low-value, mid-range, and high-value distributions, respectively.*

3) BMA in this study goes beyond a simple weighted average of model outputs. Through integration with observational data, model uncertainty, and error correction, BMA can achieve a closer alignment with actual observations. This approach allows BMA to exceed the boundary values set by individual models, especially in extreme high or low values. Our results indicate that approximately 20-30% of grid points fell outside the individual model boundaries, most notably in these extreme ranges.

Unlike simple weighted averaging, Bayesian integration could leverage observational data to update the posterior probability distribution for each model [6, 7]. This update mechanism enables BMA to produce results that deviate from conventional model averages, adjusting based on model alignment with observed data. Specifically, when there is a pronounced difference between model predictions and observed values, the Bayesian framework adapts the outcome toward the observed values through posterior updates. Observational data significantly influence the weighting of each model, resulting in iterative refinements through methods like maximum likelihood estimation, leading to a final output that better approximates observed conditions.

Additionally, BMA dynamically adjusts model weights and incorporates adjustment coefficients to account for model error and uncertainty. These adjustment coefficients apply targeted corrections to each model's output, ensuring that the integrated result is not merely an average but a refined product that reflects observational data [8, 9]. When models display significant deviations, BMA reduces their influence, favoring models that align more closely with observational data. This dynamic weighting allows the ensemble to extend beyond individual model boundaries and closely approximate true observed values, enhancing both accuracy and realism in the downscaled product.

[Figure]

Figure S4. (a) Scatter plots comparing ESA CCI soil moisture with in situ measurements from three ground station networks. (b) Scatter plots showing residual errors of the downscaled soil moisture relative to in situ measurements.

[1] Sadayappan, K., Kerins, D., Shen, C., & Li, L. (2022). Nitrate concentrations predominantly driven by human, climate, and soil properties in US rivers. *Water Research*, *226*, 119295.
[2] Bo, Y., Li, X., Liu, K., Wang, S., Li, D., Xu, Y., & Wang, M. (2024). Hybrid theory-guided data driven framework for calculating irrigation water use of three staple cereal crops in China. *Water Resources Research*, *60*(3), e2023WR035234.
[3] Wang, F., & Tian, D. (2024). Multivariate bias correction and downscaling of climate models with trend-preserving deep learning. *Climate Dynamics*, *62*(10), 9651-9672.
[4] Maraun, D., Wetterhall, F., Ireson, A. M., Chandler, R. E., Kendon, E. J., Widmann, M., ... & Thiele-Eich, I. (2010). Precipitation downscaling under climate change: Recent developments to bridge the gap between dynamical models and the end user. *Reviews of geophysics*, *48*(3).

[5] Latombe, G., Burke, A., Vrac, M., Levavasseur, G., Dumas, C., Kageyama, M., & Ramstein, G. (2018). Comparison of spatial downscaling methods of general circulation model results to study climate variability during the Last Glacial Maximum. *Geoscientific Model Development*, *11*(7), 2563-2579.

[6] Xu, T., & Valocchi, A. J. (2015). A Bayesian approach to improved calibration and prediction of groundwater models with structural error. *Water Resources Research*, *51*(11), 9290-9311.

[7] Wang, C., Wang, K., Tang, D., Hu, B., & Kelata, Y. (2022). Spatial random fields-based Bayesian method for calibrating geotechnical parameters with ground surface settlements induced by shield tunneling. *Acta Geotechnica*, *17*(4), 1503-1519.

[8] Bao, L., Gneiting, T., Grimit, E. P., Guttorp, P., & Raftery, A. E. (2010). Bias correction and Bayesian model averaging for ensemble forecasts of surface wind direction. Monthly Weather Review, 138(5), 1811-1821.

[9] Fraley, C., Raftery, A. E., & Gneiting, T. (2010). Calibrating multimodel forecast ensembles with exchangeable and missing members using Bayesian model averaging. *Monthly Weather Review*, *138*(1), 190-202.

5. The authors emphasize that one of the major advantages of their methodology is the inclusion of prior knowledge from in-situ data into the Bayesian modelling framework. However, I could not find a description of the priors they use anywhere in the paper. This needs to be included if such a strong statement is made.

It is unclear from the manuscript how the weights of the individual models in the Bayesian model averaging algorithm are derived.

Response: Thank you for raising this point. In the new version, we provide additional clarification on the use of priors and weight derivation in our BMA approach, including the role of in-situ data and our specific use of the Markov Chain Monte Carlo Model Composition ($MC^3$) algorithm. The related context can be found in the main text section 3.3.

This study utilizes BMA to create an ensemble from multiple downscaled soil moisture estimates produced by different machine learning models. BMA is a robust statistical approach that assigns weights to each model based on the posterior probability of its predictive accuracy, taking into account both model uncertainty and prior knowledge. This approach is particularly advantageous as it leverages prior information from ground-based observations (in-situ data) to refine model predictions and reduce the influence of outliers or noise in individual model outputs.

In terms of prior knowledge, we incorporate in-situ soil moisture data as a form of point-wise ground truth to guide the weighting process. Ground observations are essential in calibrating model weights by providing a reliable reference for predictive accuracy, thus allowing the ensemble model to better capture the variability and characteristics observed in the field.

1) The process of calculating the posterior probability for model weights in BMA is computationally intensive, especially when handling complex likelihood functions. To manage this, we employ the Markov Chain Monte Carlo Model Composition ($MC^3$) algorithm [1, 2], which is an efficient sampling method that approximates the posterior distribution of each model's weight.

Through $MC^3$, we iteratively sample from the model space, with each iteration using ground observation data to calibrate and refine the weight assigned to each soil moisture model. The key steps are as follows: i) Likelihood calculation with observational data: To evaluate each model's performance, we calculate the likelihood using the Negative Log-Likelihood function, which reflects the alignment between model predictions and observed data. Models with predictions that closely match observed values yield higher likelihoods, thereby increasing their probability of being assigned a higher weight during sampling. ii) Dynamic weight adjustment: During each iteration of $MC^3$, ground observations dynamically influence the weight of each model based on its alignment with the observational data. By doing so, $MC^3$ continuously optimizes the model weights, leading to a final ensemble output that is more representative of the observed field conditions. iii) Iterative sampling and optimal combination selection: Over multiple iterations, the $MC^3$ algorithm records the model combination and its likelihood, refining the weight estimates with each step. The process continues until reaching a preset sample size, after which the posterior weights for each model are derived based on cumulative likelihood values across samples.

2) The posterior probabilities (weights) calculated using $MC^3$ are initially estimated at the station scale, where ground data are available. To generalize these weights across the entire study area, we use Kriging interpolation, a spatial interpolation technique that accounts for the spatial autocorrelation of station weights. Kriging provides a way to adapt the interpolation weights to local spatial characteristics, ensuring that the ensemble weighting is appropriately distributed across regions with varying data densities and environmental conditions. Kriging also enables uncertainty estimation for the interpolated weights, which is particularly useful for assessing reliability in data-sparse areas [3]. This uncertainty information is critical for evaluating the confidence in downscaled soil moisture estimates in regions where station data are limited.

[1] Fragoso, T. M., Bertoli, W., & Louzada, F. (2018). Bayesian model averaging: A systematic review and conceptual classification. *International Statistical Review*, *86*(1), 1-28.
[2] Giudici, P., & Castelo, R. (2003). Improving Markov chain Monte Carlo model search for data mining. *Machine learning*, *50*, 127-158.
[3] Finley, A. O., Banerjee, S., & Carlin, B. P. (2007). spBayes: an R package for univariate and multivariate hierarchical point-referenced spatial models. *Journal of statistical software*, *19*(4), 1.

6. The manuscript contains misleading citations.
Response: Thank you for highlighting this issue. We have conducted a thorough review of the manuscript, with particular attention to the citations in the introduction and other sections. We have corrected all misleading or unclear citations to ensure that each reference accurately supports the statements and arguments presented.

Specific Comments
1. Introduction

The introduction lacks a clear demonstration as to why we need higher resolution soil moisture datasets (applications, hypothesis to test, etc). It would benefit from a more detailed discussion of the most influential studies that have used downscaling and what their weaknesses are. Some strong statements lack citations and some citations are misleading (see detailed comments below).

- Lines 49-51: "While these products are valuable for certain applications (Molero et al., 2016), the spatial resolution of these products—largely tens of kilometers—limits the ability to capture the spatial heterogeneity of soil moisture (Njoku and Entekhabi, 1996; Schmugge, 1998)." I am certain there are more recent studies looking into the spatial heterogeneity of soil moisture.

- Lines 52-54: "Soil moisture downscaling, an effective technique for improving spatial resolution, has received substantial attention (Zhang et al., 2022). Statistical approaches and land surface models (Famiglietti et al., 2008; Grayson and Western, 1998) have been widely used, but these methods typically require large amounts of parametric data with ground data." Famiglietti et al., 2008, does not perform any downscaling, but only provides a quantification of SM variability across scales. This citation would be better suited to the paragraph preceding this one. The last part of this sentence is also unclear. What is meant by 'parametric data with ground data'?

- Lines 54-56: "Various fusion methods integrating multi-source satellite remote sensing data have been developed, falling into categorized like active-passive microwave and optical-microwave data integration." This needs citations and it is also unclear why combining active and passive microwave sensors would increase the resolution of a dataset. It typically only increases coverage and reduces uncertainty.

- Lines 56-57: "All these mentioned models encounter challenges related to model structure constraints, data quality, scale disparities, and geographic limitations (Peng et al., 2017; Werbylo and Niemann, 2014)." Werbylo and Niemann, 2014, compares two in-situ sampling approaches for use in downscaling, but doesn't discuss challenges in downscaling.

- Lines 76-77: "Existing ensemble machine learning often overlooks the incorporation of prior knowledge, a crucial regularization mechanism that prevents overfitting and enhances model generalization." Priors can only prevent overfitting, if the overfitted solution is unlikely in prior space. A large sample size always overcomes any prior. Prior knowledge helps with small sample sizes. This needs a citation.

Response: Thank you to the reviewer for highlighting this important issue. In response, we have restructured the introduction to provide a clearer rationale for the necessity of higher-resolution soil moisture datasets. We discussed the practical applications and research hypotheses driving the need for enhanced spatial resolution, including specific applications in water resource management, drought monitoring, and precision agriculture. Additionally, we have incorporated a review of influential studies in downscaling, highlighting their achievements and limitations, to clarify the current gaps that our study aims to address. We have also thoroughly reviewed and corrected the

citations, especially for sections where stronger claims lacked supporting references, ensuring that all citations are accurate and directly relevant to the points discussed.

2. Study area and materials

1) Figure 1: Mismatch between text and figure. How do the three station types in the plot (Meteorological, Crop and CERN stations) relate to the ones mentioned in section 2.2.5 (in-situ measurements, NZW, QXZ, CERN)?

Response: Thank you for noting this discrepancy. We have revised Figure 1 to ensure consistency with the station types described in Section 2.2.5. Specifically, the figure now clearly distinguishes between the in-situ measurement types (NZW, QXZ, CERN) to align with the terminology and classification used in the text.

[Figure]

*Figure 1: Geographic Context of the Study Area. (a) Spatial distribution of land types and (b) elevation within the study area. The dots on the maps represent the precise locations of the selected ground stations employed in this research. (c) Photographic representation showcasing characteristic land types in northern China.*

2) Lines 123-125: "We use the combined active-passive ESA CCI products from 2003 to 2010, obtained from the ESA data archive (https://www.esa-soilmoisture-cci.org/)" The ESA CCI SM website clearly states that "If using the COMBINED product, the following is also compulsory in addition to the above: Preimesberger, W., Scanlon, T., Su, C. -H., Gruber, A. and Dorigo, W. (2021). Homogenization of Structural Breaks in the Global ESA CCI Soil Moisture Multisatellite Climate

Data Record, in IEEE Transactions on Geoscience and Remote Sensing, vol. 59, no. 4, pp. 2845-2862, April 2021, doi: 10.1109/TGRS.2020.3012896."
Response: Thanks for pointing out this issue. We have added this citation in the new version.

3) Line 127: "The integration of MODIS products within satellite-derived soil moisture downscaling has been extensively employed" This needs a citation.
Response: Thanks for highlighting this point. We have added an appropriate citation to support such statement in the revised manuscript.

4) Line 157: "The CERN dataset comprises 34 stations, covering a period of approximately five days from 2005 to 2014." There are only 5 measurements in 9 years? Is this correct?
Response We apologize for the initial misdescription. In the revised manuscript, we have clarified this as follows: *"The CERN dataset comprises soil moisture from 34 stations, covering the period from 2005 to 2014, with measurements taken at five-day intervals.".*

3. Methods
1) 3.1 Feasibility of chosen explanatory factors: The authors only use one (random forest) out of their 4 machine learning models for their feature importance analysis. Why this one and why not all? The results might be different for the different models. Furthermore, the authors should test for collinearity/correlation between explanatory variables to avoid overfitting their models.
Lines 208-215: The linear regression analysis doesn't add any scientific value to the paper. I would remove this paragraph along with Figure 3b.
Response: Thank you for highlighting this important point. In the revised manuscript, we have included the feature importance results directly in the main text instead of in the supplementary material. Additionally, we have expanded our analysis to address potential correlation among the explanatory variables. Specifically, we conducted a correlation analysis among the explanatory variables, finding low correlations with Pearson correlation coefficients below 0.35. This supports the model's stability by indicating minimal collinearity and reducing the risk of overfitting.

Regarding Figure 3b, we have integrated this into a broader figure illustrating autocorrelation among all relevant variables, including both satellite-derived and observed soil moisture data. This revised figure provides a clearer view of variable relationships, contributing to the overall readability and interpretability of the manuscript.

[Figure]

*Figure 3: Assessment of explanatory variables' feasibility. (a) Average (blue bar) and standard deviation (error bar) of permutation-based importance of explanatory variables concerning soil moisture. (b) Average Pearson correlation coefficients among different explanatory variables, including correlations with two independent soil moisture data sources.*

2) 3.2 Machine learning methods: it is unnecessary to explain these 4 very commonly used machine learning models.

Line 253, Equation 4: The formula here is misleading, as x should be the result of a nonlinear transformation of the explanatory variables, not the "the value of each dimension in the training set"

Response: Thanks for the reviewer's suggestion. We have removed the related context in the new version.

3) Lines 297-301: "In theory, calculating $p(M\_i|D)$ of a model involves computing the likelihood function for each model, multiplying it by the prior probability of each model, and dividing by the marginal likelihood. However, this method is rarely employed in practice due to the complexity of computing the likelihood function and prior distribution, especially for complex models with high-dimensional parameter spaces. Instead, iterative estimation techniques such as Markov Chain Monte Carlo methods are commonly used. In our study, we utilized Markov Chain Monte Carlo Cube ($MC^3$) for this purpose." Given that BMA is the main innovation in their study, the authors should explain in more detail how they calculated the individual model probabilities. Citations are needed here too.

Response: Thank you for emphasizing this point. In response, we have revised the manuscript to provide a more detailed explanation of how individual model probabilities were calculated within the BMA framework, as this is central to our study. Specifically, we have expanded the methodology section to clarify the theoretical foundation and practical implementation steps involved in the estimation process.

In our study, we employed the Markov Chain Monte Carlo Model Composition (MC³) method [1, 2], which efficiently addresses the computational complexities associated with high-dimensional parameter spaces by iteratively sampling the model space. This approach facilitates the calculation of posterior probabilities by combining each model's likelihood with its prior probability, ultimately yielding the weight of each model in the ensemble.

Additionally, we have elaborated on the role of ground observation data within the MC³ sampling process. Observational data are used to compute the likelihood values, thereby influencing model weights based on how well each model's predictions align with observed soil moisture values. This integration allows the BMA framework to adjust model weights dynamically, ensuring that the final ensemble results more accurately reflect real-world conditions.

*Specific revision: "The core idea of extending BMA from statistical models to dynamic models is that there is an optimal forecast model or member in any set of forecasts, but it is impossible to determine which model is the best. Therefore, the uncertainty of the optimal model can be quantified by BMA. The dynamic model can be used to update its weights to better reflect the variability and uncertainty in the forecasting process. Here again, y is used to denote the predicted value. Bias correction can be applied to each deterministic prediction using any of the many possible bias correction methods to generate the bias-corrected prediction $f_k$. Then the prediction $f_k$ is associated with the conditional PDF, $g_k(y|f_k)$. Assuming $f_k$ is the best prediction in the set, it can be interpreted as the conditional PDF of y conditional on $f_k$. The BMA prediction model is then:*

$$p(y|f_{1, \ldots, } f_k) = \sum_{i=1}^{k} w_k g_k(y|f_k) \tag{3}$$

*where $w_k$ is the posterior probability that prediction k is the best prediction and is based on the performance of prediction k during training. $w_k$ are probabilities, so they are non-negative and add up to 1, i.e., $\sum_{k=1}^{K} w_k = 1$.*

*In predicting remote sensing soil moisture, $g_k(y|f_k)$ can be viewed as a normally distributed density function. Its prediction mean result is a simple linear function of a single predicted result $a_k + b_k f_k$ with standard deviation σ. This can be expressed by:*

$$y|f_k \sim N(a_k + b_k f_k, \ \sigma^2) \tag{4}$$

*The above normal distribution density function is noted as:*

$$g_k(a_k + b_k f_k, \ \sigma^2) \tag{5}$$

*where: $a_k$, $b_k$ are bias correction terms that can be found by linear regression methods. In this case, the BMA predicted mean is just the conditional expectation of y given the prediction, i.e.*

$$E[y|f_{1, \ldots, } f_k] = w_k(a_k + b_k f_k) \tag{6}$$

*This prediction can be considered as a deterministic prediction. It is also possible to compare the estimated values with individual estimates or set averages in a set.*

*In this study, we used the Markov Chain Monte Carlo Model Composition (MC³) algorithm to determine the posterior probability and calculate the weights for each soil moisture product (Murray et al., 2013; Fragoso et al., 2018). Given the computational complexity involved in estimating the likelihood function and prior distribution, MC³ offered an efficient solution. Through this sampling process, ground observation data was incorporated to calibrate and refine the model weights, allowing the ensemble mean to align more closely with observed soil moisture values. This process also enables the model to extend beyond boundary values under extreme conditions, thus better capturing the variability inherent in observational data.*

*To evaluate each model's likelihood based on ground observations, we applied the Negative Log-Likelihood function, reflecting the accuracy of each model's predictions relative to observed values. Models yielding predictions that closely match observed data produce higher likelihood values, making them more likely to receive greater weights in the MC³ sampling process. The MC³ algorithm, a model combination optimization approach, efficiently samples the model space to estimate each model's posterior weight. During each iteration, the algorithm records the current model state and its likelihood, continuing this sampling process until a preset number of iterations is reached. After sampling, the posterior weights for each model are calculated. Specifically, ground observation data is central to the MC³ sampling process in three key ways: i) It is used to calculate the likelihood values, allowing models with a better fit to observed data to attain higher weights. ii) By dynamically evaluating each model's likelihood during sampling, the MC³ algorithm adjusts model weights in real time, bringing the final result closer to observed values. iii) For each sampling iteration, MC³ assesses the overall likelihood of model combinations based on ground observation data, enhancing the probability of selecting an optimal model combination. In the BMA framework, where multiple remote sensing soil moisture products are combined using ground observation data, MC³ initially calculates posterior probabilities (weights) at the station level. To generalize these station-based weights across the entire remote sensing image (non-station scale), spatial interpolation is employed. This study uses Kriging interpolation, which adapts weights based on the spatial autocorrelation characteristics of station weights, providing more accurate estimates in areas with spatial variability. Additionally, Kriging interpolation offers uncertainty estimates for the interpolated results, which are essential for evaluating data reliability, particularly in areas with sparse station data. This uncertainty information further aids in assessing the confidence level of interpolated results in data-sparse regions."*

[1] Fragoso, T. M., Bertoli, W., & Louzada, F. (2018). Bayesian model averaging: A systematic review and conceptual classification. *International Statistical Review*, *86*(1), 1-28.

[2] Giudici, P., & Castelo, R. (2003). Improving Markov chain Monte Carlo model search for data mining. *Machine learning*, *50*, 127-158.

4) Lines 324-325: "Through a sensitivity analysis conducted with an independent dataset, maximum values for these parameters are chosen for the period spanning 2003 to 2010." Which independent dataset?

Response: Thank you for highlighting this point. In our study, a sensitivity analysis was conducted using an independent dataset from the period 2011 to 2013. This analysis was instrumental in determining the optimal parameter values for model implementation over the earlier period of 2003 to 2010. Specifically, we tested various combinations of parameters, adjusting the temporal duration from 1 to 5 days (in 1-day increments) and the spatial window size from 3 to 10 units (in 1-unit increments). We have clarified this information in the revised manuscript.

4. Results and Discussions

1) The maps from Figure 5 are of too limited quality to really assess whether their method leads to improved resolution of spatial patterns in soil moisture.

Response: Thank you for your valuable feedback. In the revised version, we have refined Figure 5 to focus on several key months to provide a clearer assessment of spatial patterns.

Additionally, we have introduced the Coefficient of Variation (CV) and Moran's I index to better illustrate local and range-specific performance. These indices offer a quantitative evaluation of spatial variability, particularly highlighting an increase in variability in the mid-range of soil moisture values. It means that our approach, though conservative on extreme values, effectively captures the subtle spatial variability essential for accurate soil moisture analysis at finer resolutions. Detailed context could be found in the main text section 4.1.

[Figure]

Figure 5: Spatial distribution of soil moisture across six data sources, representing the 15th day of April-October 2009. Columns, from left to right, show the 25-km ESA CCI soil moisture and the 1-km downscaled soil moisture derived from random forest (RF), multiple linear regression (MLR), support vector regression (SVR), extreme gradient XG Boost (XG), and Bayesian model averaging (BMA), respectively.

Table Coefficient of variation (CV) and Moran's I index

|  | ESA CCI | RF | MLR | SVR | XG | BMA |
|---|---|---|---|---|---|---|
| CV | 0.321 | 0.328 | 0.313 | 0.324 | 0.332 | 0.324 |
| Moran's I [0-100%] | 0.994 | 0.964 | 0.955 | 0.994 | 0.946 | 0.972 |
| Moran's I [0-15%] | 0.991 | 0.755 | 0.876 | 0.991 | 0.846 | 0.86 |
| Moran's I [15-85%] | 0.99 | 0.795 | 0.922 | 0.989 | 0.825 | 0.803 |
| Moran's I [85-100%] | 0.991 | 0.864 | 0.921 | 0.991 | 0.895 | 0.902 |

Note: CV measures overall variability, with higher values indicating stronger heterogeneity. Moran's I quantifies spatial distribution patterns, where higher values reflect weaker heterogeneity and stronger spatial autocorrelation. The CV is calculated for the entire dataset. Moran's I index is determined using a simple four-neighborhood relationship, with brackets indicating different sample divisions. The 0-100% range represents the full sample, while the 0-15%, 15-85%, and 85-100% ranges correspond to low-value, mid-range, and high-value distributions, respectively.

2) Lines 388-390: "It is evident that the downscaled data produced by the BMA method exhibit more pronounced differences compared to the original data, particularly in terms of histogram distributions shifting towards the peak. This implies that the downscaled method effectively captures the disparities between the 25km and 1km products." Downscaled datasets should have broader distributions than the original data, as the original data should represent an average over the HR pixels contained within them. A narrowing of the distribution indicates a loss of information, rather than a gain.

Response: Thank you for highlighting this important point. In response, we have clarified the implications of the distribution changes observed in our downscaled dataset. The related context could be found in the main text section 4.1 and 4.2.

As is well known, traditional machine learning-based downscaling methods may lead to a narrower distribution due to a tendency to smooth extreme values and prioritize certain patterns in the training data, producing more conservative estimates [1, 2]. In our study, high soil moisture values made up less than 10% of the dataset. With limited representation of extreme moisture values, the model may not fully capture this range, leading it to rely predominantly on mid-to-low value ranges. This can result in underestimation at the higher end, especially in arid and semi-arid regions where high-moisture samples are sparse.

Additionally, machine learning models, such as regression and random forests, often tend toward conservative predictions to minimize overall prediction error. This can mean that extreme high values are "pulled" toward the mean, reducing variance at the higher end. During the downscaling process, the need to integrate fine-scale features such as topography and vegetation cover further complicates the capture of variability, particularly in high-moisture areas (e.g., near irrigated zones or rivers). These combined effects can lead to an underestimation of high values, while mid-range variability is effectively preserved, suggesting that the model still captures spatial heterogeneity meaningfully.

Importantly, the observed narrowing of the range should not automatically be interpreted as a loss of information [3-5]. In the context of remote sensing downscaling, a narrower range does not conflict with the enhancement of spatial variability. Increasing spatial resolution allows the model to capture finer distinctions in features such as topography, land cover, and moisture gradients. This often results in greater spatial variability across the landscape, even if the range of values is more focused. Our histograms and box plots indicate that the downscaled dataset shows more pronounced clustering around local median values, and this clustering trend intensifies post-downscaling.

Furthermore, metrics such as the Coefficient of Variation (CV) and Moran's I index confirm an increase in spatial variability, particularly in the mid-range. These metrics support the notion that, while conservative in extreme values, our downscaling method effectively captures nuanced spatial variability essential for soil moisture analysis at finer resolutions. This approach ensures

that critical regional heterogeneity is well-represented, enhancing the utility of the downscaled dataset for hydrological and environmental applications.

Table Coefficient of variation (CV) and Moran's I index

|  | ESA CCI | RF | MLR | SVR | XG | BMA |
|---|---|---|---|---|---|---|
| CV | 0.321 | 0.328 | 0.313 | 0.324 | 0.332 | 0.324 |
| Moran's I [0-100%] | 0.994 | 0.964 | 0.955 | 0.994 | 0.946 | 0.972 |
| Moran's I [0-15%] | 0.991 | 0.755 | 0.876 | 0.991 | 0.846 | 0.86 |
| Moran's I [15-85%] | 0.99 | 0.795 | 0.922 | 0.989 | 0.825 | 0.803 |
| Moran's I [85-100%] | 0.991 | 0.864 | 0.921 | 0.991 | 0.895 | 0.902 |

Note: CV measures overall variability, with higher values indicating stronger heterogeneity. Moran's I quantifies spatial distribution patterns, where higher values reflect weaker heterogeneity and stronger spatial autocorrelation. The CV is calculated for the entire dataset. Moran's I index is determined using a simple four-neighborhood relationship, with brackets indicating different sample divisions. The 0-100% range represents the full sample, while the 0-15%, 15-85%, and 85-100% ranges correspond to low-value, mid-range, and high-value distributions, respectively.

[1] Sadayappan, K., Kerins, D., Shen, C., & Li, L. (2022). Nitrate concentrations predominantly driven by human, climate, and soil properties in US rivers. *Water Research*, *226*, 119295.
[2] Bo, Y., Li, X., Liu, K., Wang, S., Li, D., Xu, Y., & Wang, M. (2024). Hybrid theory-guided data driven framework for calculating irrigation water use of three staple cereal crops in China. *Water Resources Research*, *60*(3), e2023WR035234.
[3] Wang, F., & Tian, D. (2024). Multivariate bias correction and downscaling of climate models with trend-preserving deep learning. *Climate Dynamics*, *62*(10), 9651-9672.
[4] Maraun, D., Wetterhall, F., Ireson, A. M., Chandler, R. E., Kendon, E. J., Widmann, M., ... & Thiele-Eich, I. (2010). Precipitation downscaling under climate change: Recent developments to bridge the gap between dynamical models and the end user. *Reviews of geophysics*, *48*(3).
[5] Latombe, G., Burke, A., Vrac, M., Levavasseur, G., Dumas, C., Kageyama, M., & Ramstein, G. (2018). Comparison of spatial downscaling methods of general circulation model results to study climate variability during the Last Glacial Maximum. *Geoscientific Model Development*, *11*(7), 2563-2579.

3) Lines 397-398: "While most of the downscaling results exhibit lower values compared to the original ESA CCI values during most months, this variance is not of substantial magnitude." This is not a variance, but a bias.
Response: Thank you for highlighting this distinction. We have revised the statement in the new version to accurately refer to this as a "bias" rather than "variance".

4) Line 398-399: "This pattern can be attributed to the inherent characteristics of the BMA ensemble approach, which combines multiple machine learning outcomes to prevent excessively high or low values." This might explain the lower variance, but not the bias.

Response: We appreciate your guidance, which has helped us clarify these aspects in the manuscript. In the revised version, we have removed this statement to avoid potential confusion. Additionally, we have provided a more detailed explanation of the lower variance observed in the results in Section 4.2 and 4.3, where we discuss the inherent properties of the BMA ensemble approach.

5) Lines 419-420: "The absence of in situ data in the western desert-dominated region potentially weak the model training, exerting a negative effect on the resultant model accuracy." Until this point, we don't know how the in-situ data is used in training. I assume it enters the model probabilities in the BMA, but this is never explained.

Response: Thank you for pointing out the need for clarity on the use of in-situ data within the BMA framework. We have now expanded our explanation to describe how in-situ soil moisture data is used in the model training and weighting process. The related context could be found in the main text section 3.3.

In this study, we incorporate in-situ soil moisture observations as point-specific ground truth data to refine and calibrate model weights, which enhances the ensemble's alignment with real-world soil moisture variability. This in-situ data is essential for ensuring the model's ability to accurately capture soil moisture patterns and to mitigate any biases due to limited data coverage in specific regions, such as the western desert areas.

The weighting process in BMA is accomplished through the Markov Chain Monte Carlo Model Composition ($MC^3$) algorithm, an efficient sampling approach that approximates the posterior distribution of each model's weight. This iterative process utilizes ground observations to adjust the weight of each model based on its predictive alignment with observed data. The main steps are as follows: i) likelihood calculation with observational data: to gauge each model's predictive accuracy, we calculate a likelihood measure using the Negative Log-Likelihood function, comparing model predictions to observed soil moisture values. Models that closely match these observations yield higher likelihood scores, which in turn increases their probability of receiving a higher weight during the sampling process. ii) dynamic weight adjustment: during each iteration of the $MC^3$ process, the algorithm recalibrates model weights based on the observed alignment with in-situ data. This dynamic adjustment allows ground observations to continually influence model weights, making each iteration progressively more representative of field conditions. iii) iterative sampling and optimal combination selection: over multiple iterations, $MC^3$ records the model combinations and their likelihood scores, refining weight estimates with each cycle. The sampling continues until a preset sample size is reached, allowing for a stable estimation of the posterior weights based on cumulative likelihood values from the entire sampling process. Collectively, our framework allows the ensemble to achieve a closer alignment with observed field conditions, as the in-situ data informs each model's contribution based on its performance relative to ground truth data.

6) Figure 7: colour scales are confusing, high MAE should correspond to low R (and thus have the same colours)

Response: Thank you for highlighting this important point regarding the color scales in Figure 7. We have now adjusted the color scale to ensure consistency, with high Mean Absolute Error (MAE) values and low correlation (R) values aligned in similar colors to reflect the expected inverse relationship.

[Figure]

*Figure 7: Performance evaluation of downscaled soil moisture. (a) Correlation coefficient (R) and (b) mean absolute error (MAE) illustrating the accuracy of BMA-based downscaled soil moisture when compared against in situ measurements from three ground station networks.*

7) Lines 431-439: The authors should include confidence intervals for these metrics. Furthermore, I wonder how they compare to their coarse-grained modelled dataset? I assume they would be very similar, which would suggest that the downscaling has little effect, but that the results rather stem from the models smoothing the data (at any scale).

Response: Thank you for your valuable comments. We have analyzed the confidence intervals for our metrics to check the reliability of our results. Our analysis of confidence levels in the correlation validation results revealed that only the CERN station achieved a significance level

below 0.05, while the other two stations did not display statistically significant confidence levels. However, we recognize that the confidence level of correlation between ground station data and downscaling results may not fully validate model performance alone, highlighting in several other studies [1-2], as accuracy is also influenced by factors such as spatial scale mismatches, error propagation, and inherent uncertainties. This is the reason why most current studies tend to incorporate additional metrics beyond correlation analysis, specifically Root Mean Square Error (RMSE) and Mean Absolute Error (MAE), which offer a fuller picture of model accuracy.

Additionally, we compared downscaled results with the original coarse-resolution remote sensing data and observed notable differences. The inclusion of high-resolution explanatory variables contributed substantially to improving accuracy and capturing heterogeneity, particularly on local scales and within mid-range values. Beyond the two heterogeneity metrics discussed earlier, we also assessed the accuracy within the middle range of values [15–85%], where we observed a significant improvement compared to overall accuracy. These localized metrics effectively illustrate the impact of our algorithm, suggesting that the downscaling approach enhances both spatial resolution and reliability across the dataset.

In the new version, we have added the related context could be found in the main text section 4.3 and supplementary Fig. S5.

[Figure]

*Figure S5. Correlation between 1-km downscaled soil moisture and the original 25-km satellite-derived soil moisture, including the difference between the two datasets.*

*Table 4. Comparison of BMA and individual machine learning*

| Stations | Num | R | | | | | | | | | | | |
|---|---|---|---|---|---|---|---|---|---|---|---|---|---|
| | | RF | | MLR | | SVR | | XG | | BMA | | ESA CCI | |
| | | All | Mid | All | Mid | All | Mid | All | Mid | All | Mid | All | Mid |
| NZW | 5044 | 0.323 | 0.383 | 0.338 | 0.411 | 0.321 | 0.398 | 0.325 | 0.399 | 0.342 | 0.424 | 0.321 | 0.375 |
| CERN | 263 | 0.567 | 0.664 | 0.617 | 0.693 | 0.629 | 0.705 | 0.573 | 0.672 | 0.642 | 0.721 | 0.586 | 0.647 |
| QXZ | 1204 | 0.477 | 0.571 | 0.480 | 0.593 | 0.478 | 0.583 | 0.468 | 0.554 | 0.514 | 0.610 | 0.479 | 0.531 |
| | | RMSE (m3/m3) | | | | | | | | | | | |
| Stations | Num | RF | | MLR | | SVR | | XG | | BMA | | ESA CCI | |
| | | All | Mid | All | Mid | All | Mid | All | Mid | All | Mid | All | Mid |

| Stations | Num | | | | | | | | | | | | |
|---|---|---|---|---|---|---|---|---|---|---|---|---|---|
| NZW | 5044 | 0.138 | 0.108 | 0.136 | 0.105 | 0.138 | 0.106 | 0.137 | 0.104 | 0.137 | 0.097 | 0.138 | 0.115 |
| CERN | 263 | 0.073 | 0.060 | 0.073 | 0.059 | 0.073 | 0.060 | 0.075 | 0.061 | 0.071 | 0.054 | 0.084 | 0.064 |
| QXZ | 1204 | 0.169 | 0.125 | 0.169 | 0.131 | 0.168 | 0.132 | 0.169 | 0.128 | 0.169 | 0.115 | 0.168 | 0.139 |

| Stations | Num | MAE (m3/m3) | | | | | | | | | | | |
|---|---|---|---|---|---|---|---|---|---|---|---|---|---|
| | | RF | | MLR | | SVR | | XG | | BMA | | ESA CCI | |
| | | All | Mid | All | Mid | All | Mid | All | Mid | All | Mid | All | Mid |
| NZW | 5044 | 0.114 | 0.092 | 0.113 | 0.091 | 0.114 | 0.095 | 0.114 | 0.095 | 0.114 | 0.089 | 0.114 | 0.101 |
| CERN | 263 | 0.060 | 0.038 | 0.061 | 0.039 | 0.061 | 0.039 | 0.062 | 0.040 | 0.058 | 0.032 | 0.067 | 0.045 |
| QXZ | 1204 | 0.157 | 0.119 | 0.158 | 0.123 | 0.156 | 0.117 | 0.158 | 0.120 | 0.158 | 0.114 | 0.157 | 0.134 |

*Note: "All" refers to the full set of sample points, whereas "Mid" denotes the subset of sample points that fall within the 15-85% range.*

[1] Crow W T, Van den Berg M J. An improved approach for estimating observation and model error parameters in soil moisture data assimilation. Water Resources Research, 2010, 46(12).
[2] Miralles D G, De Jeu R A M, Gash J H, et al. Magnitude and variability of land evaporation and its components at the global scale. Hydrology and Earth System Sciences, 2011, 15(3): 967-981.

8) Lines 440-450: I don't see how this paragraph is relevant to the paper. Accurately predicting soil moisture during Monsoon season is not a question of downscaling. The authors also never mention a particular focus on Monsoon prediction in the introduction or elsewhere.
Response: Thank you for your valuable feedback. Based on input from multiple reviewers, we have removed the references to Monsoon season predictions, as they were not directly relevant to the scope and focus of this study. Instead, we have added a more detailed analysis of soil moisture accuracy and heterogeneity across various ranges, as the above Table 4. This additional information enhances the evaluation of our model's performance and provides a more comprehensive assessment of its effectiveness in different soil moisture conditions.

9) In Figure 9, the time-series values from the BMA are either consistently higher or lower than all of the underlying model predictions. A weighted average must lie somewhere between its constituent values! This suggests that the authors are making an error in their calculations.
Response: Thank you for this insightful observation. The BMA approach used in our study is more than a straightforward weighted average of model outputs. By integrating observational data, accounting for model uncertainty, and incorporating error correction, BMA can align more closely with real-world observations. This approach allows BMA to produce values that may occasionally fall outside the range of the individual model predictions, particularly for extreme high or low values. The selected stations in Figure 9, which represent relatively arid or humid conditions, help illustrate this behavior.

Unlike traditional weighted averaging, Bayesian integration utilizes observational data to update the posterior probability distributions for each model [1, 2]. This mechanism allows BMA

to adjust results beyond typical model averages, favoring values that align with observed data. Specifically, when there is a marked difference between model outputs and observations, the Bayesian framework shifts the final estimate toward the observed values through these posterior updates. Observational data play a significant role in recalibrating model weights iteratively, often using techniques such as maximum likelihood estimation to produce an output that more accurately reflects the actual environmental conditions.

Moreover, BMA dynamically adjusts model weights with correction coefficients that account for model errors and uncertainties [3, 4]. These coefficients apply focused adjustments to each model's output, so the final integrated result is a refined estimate rather than a simple average. When models exhibit considerable deviations from observed values, BMA down-weights these models and increases the influence of those models that align more closely with observations. This process allows the ensemble to extend beyond the boundary values of individual models, ultimately generating a more realistic and accurate downscaled product that closely mirrors true soil moisture conditions.

In the new version, we have added the related context could be found in the main text section 4.3.

[1] Xu, T., & Valocchi, A. J. (2015). A Bayesian approach to improved calibration and prediction of groundwater models with structural error. *Water Resources Research*, *51*(11), 9290-9311.
[2] Wang, C., Wang, K., Tang, D., Hu, B., & Kelata, Y. (2022). Spatial random fields-based Bayesian method for calibrating geotechnical parameters with ground surface settlements induced by shield tunneling. *Acta Geotechnica*, *17*(4), 1503-1519.
[3] Bao, L., Gneiting, T., Grimit, E. P., Guttorp, P., & Raftery, A. E. (2010). Bias correction and Bayesian model averaging for ensemble forecasts of surface wind direction. Monthly Weather Review, 138(5), 1811-1821.
[4] Fraley, C., Raftery, A. E., & Gneiting, T. (2010). Calibrating multimodel forecast ensembles with exchangeable and missing members using Bayesian model averaging. *Monthly Weather Review*, *138*(1), 190-202.

10) Lines 488-489: "As illustrated in Fig. 10, the R and MAE distributions of the ERA5 data within the study area and the Noah data within the Loess Plateau are utilized." Why is the Noah data not used for the whole study area too? Unless a reason for this is given, it looks like cherry-picking.
Response: Thank you for bringing up this important point. We recognize that limiting the use of the Noah model to only a portion of the study area could appear selective without further clarification. The decision to use the Noah model only for the Loess Plateau region, rather than the entire study area, was driven by the model's high computational demands, which require significant resources. Given the time and budget constraints of this research, running the Noah model across the entire study area would not have been feasible [1]. To address these constraints while meeting our research objectives, we applied the Noah model specifically to the Loess Plateau,

a key area of interest, and used the more computationally efficient ERA5 data for the surrounding regions.

The Loess Plateau was selected as the focal area for the Noah model due to its unique soil structure and hydrological characteristics [2]. This region also encompasses typical semi-arid and arid landscapes, including forests and agricultural land, making it highly representative and relevant to the study's goals. We believe that applying the Noah model in this specific region allows us to capture finer-scale soil and hydrological dynamics with greater accuracy. In contrast, using ERA5 data in other regions achieves an acceptable balance between computational demand and overall data representativeness, ensuring that our findings remain robust and applicable across the broader study area.

In the new version, we have added the related context could be found in the main text section 2.2.6 and 3.5.

[1] Niu G Y, Yang Z L, Mitchell K E, et al. The community Noah land surface model with multiparameterization options (Noah-MP): 1. Model description and evaluation with local-scale measurements[J]. Journal of Geophysical Research: Atmospheres, 2011, 116(D12).
[2] Liu K, Li X, Wang S, et al. Unrevealing past and future vegetation restoration on the Loess Plateau and its impact on terrestrial water storage[J]. Journal of Hydrology, 2023, 617: 129021.

11) Lines 489-491: "Results reveal that the BMA ensemble outcomes exhibit reasonable performance in terms of higher R values and lower MAE values when compared to both the ERA5 and Noah datasets." Higher R and lower MAE compared to what? The original CCI SM dataset? Please specify.
Response: Thank you for pointing out this need for clarification. In the revised version, we have specified the comparison as follows: *"Results reveal that the BMA ensemble outcomes exhibit reasonable performance in terms of higher R values and lower MAE values when evaluated against the ERA5 and Noah datasets as reference standards, respectively."*

12) Section 4.6 Uncertainty analysis: this paragraph does not constitute uncertainty analysis, but a feature importance analysis.
Response: Thank you for highlighting this distinction. After reviewing this section alongside issue #14, we have revised Sections 4.5 and 4.6 by combining them into a single, cohesive section now titled "Model performance assessment". This section now appropriately reflects the focus on feature importance and model performance analysis.

13) Table 5: The authors should not mix R and R², but rather pick one and transform the other. They should report the overall R, rather than the range over in-situ networks. This is misleading.
Response: Thank you for your valuable feedback. In the revised version, we have standardized the metric to report only the correlation coefficient, "R," throughout the table for consistency. It should be clarified that the range shown for evaluation metrics represents calculations made individually

for each in-situ network within the study area, rather than an overall metric across all networks. We have added this in the new version.

14) Figures S6 and S7 (comparing the model performance with and without clustering and the spatiotemporal searching window) should be incorporated into the main manuscript, as they highlight the superiority of their novel approach. Figures 11 and 12 could move to the supporting information as they don't add much value to the paper.

Response: Thank you for the reviewer's suggestion. Based on the reviewer's comments, we have moved Figures S6 and S7 into the main manuscript and combined them with Figure 11. Additionally, we have moved Figure 11 to the supporting information.